# Quantized Berry winding from an emergent $\mathcal{PT}$ symmetry

Thibaud Louvet[1], Pierre Delplace[2], Mark Oliver Goerbig[3], David Carpentier[2]

**1** Université de Lyon, Institut des Nanotechnologies de Lyon, INL-UMR5270, CNRS, Ecole Centrale de Lyon, 36 avenue Guy de Collongue, Ecully, F-69134, France
**2** ENS de Lyon, CNRS, Laboratoire de Physique, F-69342, Lyon.
**3** Laboratoire de Physique des Solides, CNRS UMR 8502, Université Paris-Saclay, F-91405 Orsay Cedex, France
* thibaud.louvet@ec-lyon.fr

November 29, 2021

## Abstract

**Linear crossing of energy bands occur in a wide variety of materials. In this paper we address the question of the quantization of the Berry winding characterizing the topology of these crossings in dimension $D = 2$. Based on the historical example of 2-bands crossing occuring in graphene, we propose to relate these Berry windings to the topological Chern number of a $D = 3$ dimensional extension of these crossings. This dimensional embedding is obtained through a choice of a gap-opening potential. We show that the presence of an (emergent) $\mathcal{PT}$ symmetry, local in momentum and antiunitary, allows us to relate $D = 3$ Chern numbers to $D = 2$ Berry windings quantized as multiple of $\pi$. We illustrate this quantization mechanism on a variety of three-band crossings.**

# 1   Introduction

In recent decades, topology has allowed us to deepen our understanding of various band structures. While initially applied to gapped phases [1], be they insulators or superconductors, topological tools have then been extended to various other band structures, including those with band crossings. In the historical example of the two-dimensional (2D) graphene [2], two bands cross linearly. This crossing can be characterized by a topological number, *i.e.* a quantity robust to smooth deformations of the Hamiltonian which preserve the crossing. In this case, the topological index is a Berry winding : the phase acquired by an eigenstate smoothly wound around the band crossing point in momentum space, which takes a value $\pi$ in graphene. Such a Berry winding plays an important role in physical quantities : it manifests itself in the dispersion of Landau levels and thus in the quantum Hall effect in graphene, which is indeed characterized by an anomalous conductance quantization [3]. Various topological indices have been proposed to characterize band crossing points beyond those of graphene, such as 3D Dirac four-band crossing [4,5] and Weyl two-band crossings [6–8]. Symmetry-enforced topological invariants have been identified to characterize these band crossings [9–13] .

In this paper, we develop an alternative description of such topological properties for two- and three-band crossings in dimension $D = 2$. Our approach is based on a description of the local band crossing in momentum space, applicable both in solids and in continuous media, irrespective of any crystalline symmetries. Our strategy consists in first immersing the crossing into dimension $D = 3$ by introducing a gap opening mass term, the mass playing the role of the third dimension. We then relate the Berry winding of a given band along a path around the band crossing point in $D = 2$ to the Chern number of the corresponding band on a surface enclosing the band crossing point in $D = 3$ – that we dub 3D Chern number. We argue that this Berry winding is quantized provided an antiunitary $\mathcal{PT}$ symmetry emerges at low energy. We do not require a global $\mathcal{PT}$ symmetry, but only an emergent symmetry that is local in reciprocal space. This mechanism takes its origin in the canonical example of two-band crossing occurring *e.g.* in graphene. In this case, the presence of the actual parity

and time-reversal symmetry ensures that their combination $\mathcal{PT}$ preserves the stability of the band crossing at Dirac points. We illustrate this mechanism on various examples of three-band crossings.

# 2 From a Berry winding in $D = 2$ to $D = 3$ Chern number *via* a $\mathcal{PT}$ symmetry.

## 2.1 Berry winding around a 2-band crossing

We start the presentation with a discussion of a two-band crossing in 2D, described by the generic massless Dirac Hamiltonian

$$H^{2D}(\mathbf{k}) = k_x \sigma_x + k_y \sigma_y \tag{1}$$

where $\sigma_x, \sigma_y$ represent the $x$ and $y$ Pauli matrices, and the momentum $\mathbf{k} = (k_x, k_y)$ is relative to the band crossing point which occurs at $\mathbf{k} = \mathbf{0}$. Such a Hamiltonian describes $k$-locally[1] *e.g.* the band structure close to one of the valleys in graphene, or along a critical line between two distinct topological phases of the Haldane model [14] where the gap closes at a single point in momentum space.

The geometry of the eigenstates $|\psi_\pm\rangle$ of the two bands $\varepsilon_\pm = \pm|\mathbf{k}|$ of model (1) is captured by the associated Berry connections

$$\mathbf{A}_\pm = -\mathrm{i}\langle\psi_\pm|\boldsymbol{\nabla}_\mathbf{k}|\psi_\pm\rangle = \pm\frac{1}{2}\boldsymbol{\nabla}_\mathbf{k}\varphi \tag{2}$$

where we have introduced the polar coordinates $\mathbf{k} = (k, \varphi)$. The Berry connections (2) play a role analogous to a magnetic potential in momentum space. In particular, degeneracy points act as sources of *Berry flux* in momentum space. Analogous to the Aharonov-Bohm phase around an electromagnetic flux tube, the winding of the Berry connections along a close path $\mathcal{C}$ that encircles clockwise the degeneracy point (1) yields a phase

$$\gamma_\pm = \int_\mathcal{C} \mathbf{A}_\pm d\mathbf{k} = \frac{1}{2}\int_0^{2\pi} d\varphi = \pm\pi. \tag{3}$$

We shall refer to this local topological property of the band crossing as the *Berry winding* in the following. On a technical side, let us note that the above definition assumes that the Bloch Hamiltonians are written in the unit-cell convention [12] or convention I [15]. The Berry windings $\gamma$ we consider are related to the Wilson loop along a non-contractible loop $\ell$ that encircles the band crossing, see appendix A.

In this paper, we address the question of the quantization of this Berry winding in units of $\pi$. Given that (3) is a local property of 2D two-band crossings, we want to resort to local constraints in momentum space, as opposed to crystalline symmetries. The general strategy we will follow is to relate it to the first Chern number of eigenstates obtained upon dimensional extension to $D = 3$ by introducing a mass term. The Berry winding in $D = 2$ will acquire quantization from the topological Chern number in $D = 3$ provided an effective antiunitary $k$-local $\mathcal{PT}$ symmetry emerges at low energy. We will illustrate this mechanism on the simplest two-band crossing (1) in the following section.

---

[1] We remind the reader that we consider locality in reciprocal space here, whence the term "$k$-locality".

## 2.2  Embedding a 2D Dirac point in 3D parameter space : from Berry winding to Chern number

A generic mass opening potential that lifts the band crossing degeneracy of the Hamiltonian (1) leads to the Hamiltonian

$$H^{3D} = k_x \sigma_x + k_y \sigma_y + m \sigma_z \ . \tag{4}$$

Such a mass term opens a gap in the dispersion relation $\epsilon_\pm(\mathbf{k}) = \pm\sqrt{\mathbf{k}^2 + m^2}$, which is generically parametrized by 3 parameters $\mathbf{p} = (k_x, k_y, m)$ in the Brillouin Zone $\times$ Mass space. The continuous variation of the mass parameter provides us thus with an additional dimension. This Hamiltonian is formally equivalent to a 3D Weyl Hamiltonian. In spherical coordinates $\mathbf{p} = (k_x, k_y, m) = p(\sin\theta \ \cos\varphi, \sin\theta \ \sin\varphi, \cos\theta)$, the normalized eigenstates actually only depend on the angular direction from the degeneracy point, such as

$$\psi_+^N(\mathbf{k}, m) = \begin{pmatrix} \cos\frac{\theta}{2} \\ \sin\frac{\theta}{2} \ e^{i\varphi} \end{pmatrix}, \ \psi_-^N(\mathbf{k}, m) = \begin{pmatrix} \sin\frac{\theta}{2} \ e^{-i\varphi} \\ -\cos\frac{\theta}{2} \end{pmatrix} \ . \tag{5}$$

Those eigenstates are well defined everywhere in parameter space except on the negative $m$ axis where $\theta = \pi$, where they are multi-valued. Indeed, the Berry monopole at $\mathbf{p} = \mathbf{0}$

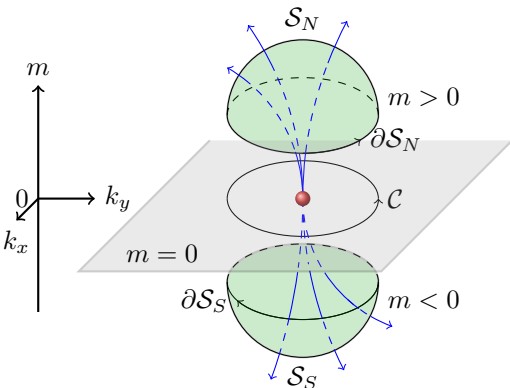

Figure 1:  A 2D band crossing in momentum space $(k_x, k_y)$ can be embedded in a $D = 3$ space with the additional dimensional provided by the amplitude $m$ of a gap opening mass term. When an antiunitary transformation of the 3D Hamiltonian relates the northern $(m > 0)$ and southern $(m < 0)$ sides of the plane, the Berry winding along a closed loop $\mathcal{C}$ in the $(k_x, k_y)$ plane can be related to the integral of the Berry flux on a closed surface $\mathcal{S}_S \mathrm{U} \mathcal{S}_N$ encircling the crossing, a Chern number.

acts as an obstruction to smoothly define smoothly eigenstates in every direction around the degeneracy point in 3D. The eigenstates can only be piece-wise defined to cover the whole $\mathbf{p}$-sphere. This is achieved by using a second gauge well-defined on the negative-$m$ axis, *i.e.* the South hemisphere $\mathcal{S}_S$:

$$\psi_\pm^S = e^{\mp i\varphi} \psi_\pm^N \ . \tag{6}$$

Accordingly, the Berry connections for the band $\varepsilon_+$ associated with the different gauge choices $\psi_+^N$ and $\psi_+^S$ read

$$\mathbf{A}_+^N = -\mathrm{Im}\langle\psi_+^N|\boldsymbol{\nabla}_\mathbf{p}|\psi_+^N\rangle = \sin^2\left(\frac{\theta}{2}\right)\boldsymbol{\nabla}_\mathbf{p}\varphi, \tag{7a}$$

$$\mathbf{A}_+^S = -\mathrm{Im}\langle\psi_+^S|\boldsymbol{\nabla}_\mathbf{p}|\psi_+^S\rangle = -\cos^2\left(\frac{\theta}{2}\right)\boldsymbol{\nabla}_\mathbf{p}\varphi. \tag{7b}$$

On the equator plane $m = 0$, the two connections are well defined and are related by a gauge transformation:

$$\mathbf{A}_+^N = \mathbf{A}_+^S + \boldsymbol{\nabla}_\mathbf{p}\varphi \ . \tag{8}$$

The impossibility to find a single smooth global phase for the eigenstates everywhere around the band crossing point is a topological property of these eigenstates $|\psi_+(\mathbf{p})\rangle$ defined over $\mathbb{R}^3\backslash\{0\}$, insensitive to any smooth deformation of these vectors. The presence of this obstruction is encoded into the first Chern number $\nu_\pm$. This integer-valued topological index can be expressed as the net flux of Berry curvature $\mathbf{F}_\pm = \boldsymbol{\nabla}\times\mathbf{A}_\pm$ emanating from the degeneracy point through a closed surface $\mathcal{S}$ enclosing the origin in parameter space :

$$\nu_\pm = \frac{1}{2\pi}\oiint_\mathcal{S}\mathbf{F}_\pm\cdot d\mathbf{S} \tag{9}$$

The Berry curvature being insensitive of the gauge choice for $\psi_\pm$, a straightforward calculation of the Chern number consists in splitting the closed surface surrounding the Berry monopole into two hemispheres $\mathcal{S}_N$ and $\mathcal{S}_S$ over which the eigenstates can be smoothly defined in the appropriate gauges (see Fig. 2). Then, using Stokes theorem, the surface integral of the Berry curvature is reduced to two line integrals of the Berry connections $\mathbf{A}_N$ and $\mathbf{A}_S$ encircling the degeneracy point in the equatorial plane $(m = 0)$ as

$$\nu_+ = \frac{1}{2\pi}\left(\oint_{\partial\mathcal{S}_N}\mathbf{A}_+^N\cdot d\boldsymbol{\ell} + \oint_{\partial\mathcal{S}_S}\mathbf{A}_+^S\cdot d\boldsymbol{\ell}\right) \tag{10}$$

$$= \frac{1}{2\pi}\oint_{\mathcal{C}=\partial\mathcal{S}_N}(\mathbf{A}_+^N - \mathbf{A}_+^S)\cdot d\boldsymbol{\ell} \tag{11}$$

where $\partial\mathcal{S}_N$ and $\partial\mathcal{S}_S$ denote the boundaries of the North and South hemispheres, respectively. By inserting the relation (8), the Chern numbers of the band $\varepsilon_+$ reads

$$\nu_+ = \frac{1}{2\pi}\int_0^{2\pi}d\varphi = 1 \ . \tag{12}$$

We have computed the Chern number associated to the two-band crossing point as an integral of the Berry flux threading a surface enclosing the degeneracy point (at the origin) in 3D $\mathbf{p}$-space. In the course of the derivation, we have seen that this chern number reduces to the difference (12) between the integrals of the different Berry connections (7a) and (7b) along the equator, i.e. on a closed loop encircling the origin in the $m = 0$ ($\theta = \pi/2$) plane, see Fig. 1. Evaluating the connections (7a) on the $m = 0$ ($\theta = \pi/2$) plane, we find

$$\mathbf{A}_+^N(m=0) = -\mathbf{A}_+^S(m=0) = \frac{1}{2}\boldsymbol{\nabla}\varphi = \mathbf{A}_+^{2D}, \tag{13}$$

where $\mathbf{A}_+^{2D}$ is the 2D Berry connection (2). We therefore have the following relation between the Chern number (12) of the Berry monopole in 3D $\mathbf{p}$-space and the Berry winding (3) of the same degeneracy point in the 2D plane at $m = 0$:

$$\nu_+ = \frac{1}{2\pi} \oint_{\partial \mathcal{S}_N} (\mathbf{A}_+^N - \mathbf{A}_+^S) \cdot d\boldsymbol{\ell} = \frac{1}{2\pi} \oint_{\partial \mathcal{S}_N} 2\mathbf{A}_+^{2D} \cdot d\boldsymbol{\ell} = \frac{1}{\pi} \gamma_+. \tag{14}$$

This expression relates the $\pi$-quantization of the Berry winding around the Dirac point in $D = 2$ $(k_x, k_y)$ momentum space to the value of the corresponding Chern number in $D = 3$ $(k_x, k_y, m)$ parameter space. Let us now explore the origin of this relation.

## 2.3 Effective antiunitary $\mathcal{PT}$ symmetry and Berry winding quantization

The correspondence (14) between a Chern number in $(k_x, k_y, m)$ space and a *quantized* Berry winding in $(k_x, k_y)$ space follows from the relation (13) between the Berry connections (7a) and (7b) at the equator. This relation is guaranteed by the existence of an antiunitary symmetry of the Hamiltonian $H^{3D}$

$$V^{-1} H^{3D}(k_x, k_y, m) V = \sigma_x (H^{3D})^*(k_x, k_y, m) \sigma_x = H^{3D}(k_x, k_y, -m). \tag{15}$$

where $V$ is an antiunitary operator, which, for (4), reads simply

$$V = \sigma_x \mathcal{K}, \tag{16}$$

with the complex conjugation $\mathcal{K}$. This transformation relates eigenstates of same momentum $\mathbf{k}$, but in different $U(1)$ gauges related to two opposite mass terms $m$, such as

$$\psi_+^N(\mathbf{k}, m) = \begin{pmatrix} \cos\frac{\theta}{2} \\ \sin\frac{\theta}{2}\, e^{i\varphi} \end{pmatrix} = V \begin{pmatrix} \sin\frac{\theta}{2}\, e^{-i\varphi} \\ \cos\frac{\theta}{2} \end{pmatrix} = V \psi_+^S(\mathbf{k}, -m) . \tag{17}$$

On the equator $(\theta = \pi/2)$, $V$ thus acts as an effective $\mathcal{PT}$ symmetry for the Hamiltonian $H^{3D}(m = 0) = H^{2D}$, as it is local in $\mathbf{k}$ and antiunitary. This $\mathcal{PT}$ symmetry relates now the eigenstates of $H^{2D}$ expressed in different gauges as

$$\psi_+^N(m = 0) = V \psi_+^S(m = 0) \tag{18}$$

which, in turns, yields a relation between the Berry connections in the North and South gauges (7a) and (7b) at the equator

$$\begin{aligned} \mathbf{A}_+^N(m = 0) &= \mathrm{Im}\langle V\psi_+^S(m = 0)|\boldsymbol{\nabla}_\mathbf{k}|V\psi_+^S(m = 0)\rangle \\ &= \mathrm{Im}\left(\langle \psi_+^S(m = 0)|\boldsymbol{\nabla}_\mathbf{k}|\psi_+^S(m = 0)\rangle\right)^* \\ &= -\mathbf{A}_+^S(m = 0). \end{aligned} \tag{19}$$

This is precisely the relation (13) at the origin of the Berry winding quantization. This unveils the role of a $\mathcal{PT}$ symmetry in 2D that is broken in 3D by the mass term and that maps North and South eigenstates to ensure the quantization of the Berry winding. Let us note that, in Ref. [12], a quantization condition similar to (18) is deduced from space-time inversion ($\mathcal{PT}$) symmetry in the context of Topological Crystalline Insulators. However, the $\mathcal{PT}$ symmetry is local in momentum space as opposed to standard crystalline symmetries and therefore applies *e.g.* in the absence of lattice structures. Here we show that an effective $\mathcal{PT}$ symmetry is a sufficient condition for the quantization, even in absence of standard crystalline symmetries. This applies therefore in systems such as continuous media.

## 2.4 $SU(2)$ rotation and Dirac strings

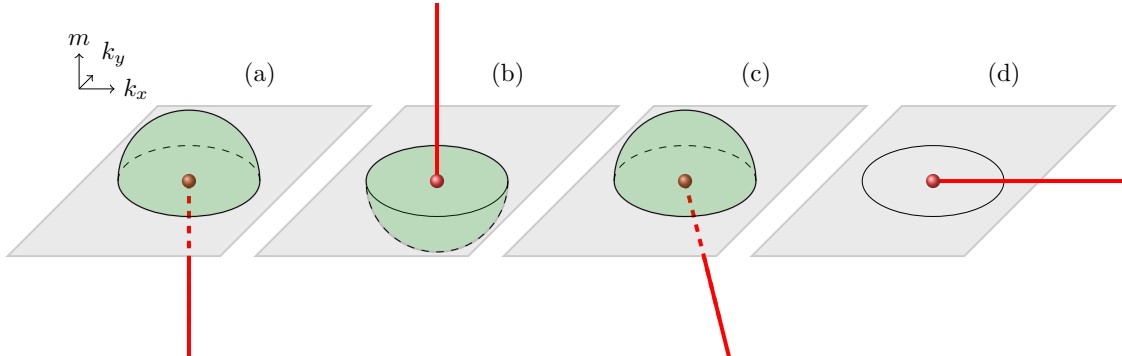

Figure 2: The Berry connection describes a monopole of Berry flux at the origin in **p** space together with a tube of flux corresponding to the so-called Dirac string in the case of the Dirac monopole of electromagnetics. The orientation of the Dirac string is gauge dependant. More precisely, its direction depends on the choice of basis representation for the Hilbert space describing the crossing. Thus in the case of the 2 band crossing in $d = 2$, the orientation of the Dirac string corresponds to a spin axis of quantization. Along this axis, the orientation of the Dirac string is $U(1)$ gauge dependent. Different choices are discussed. a) For the so-called North gauge, the Dirac string points along the $m < 0$ axis, and the eigenstates are smoothly defined for any $m \geq 0$. b) For the so-called South gauge, the situation is symmetric, with eigenstates smoothly defined for any $m \leq 0$ and a Dirac string pointing along the $m > 0$ axis. c) Arbitrary gauge, with a Dirac string located in the $m < 0$ half space, and thus eigenstates smoothly defined for $m \geq 0$. d) A pathological case where the Dirac string is in the equatorial plane. This gauge choice leads to an obstruction to the definition of smoothly defined eigenstates for the $D = 2$ model for $m = 0$. For this gauge choice, the discontinuity of the eigenstates along the Dirac string contributes to the Berry winding.

The Berry connection (7a) describes a Berry monopole in 3D parameter space $(k_x, k_y, m)$. This monopole is associated with a half flux line, that is given by the half line in the direction where the connection is ill-defined. This half-line is reminiscent of the *Dirac string* of magnetic monopoles, described with vector potentials that are ill-defined along semi-infinite line in space, around which the circulation of the vector potential yields a finite magnetic flux. Consider, for example, the connection $\mathbf{A}^N$. It is ill-defined when $\theta = \pi$, which unveils a Dirac string along the semi-infinite line $m < 0$. Indeed, the circulation of the connection around this line over a radius $k \sin\theta$ at polar angle $\theta$ gives

$$\oint_\theta \mathbf{A}_+^N d\mathbf{k} = \oint_\theta \sin^2 \frac{\theta}{2} \boldsymbol{\nabla}_\mathbf{k}\varphi \, d\mathbf{k} = \sin^2 \frac{\theta}{2} \int_0^{2\pi} d\varphi = 2\pi \sin^2 \frac{\theta}{2}. \tag{20}$$

For a vanishing radius, *i.e.* when closing the angle, it gives:

$$\lim_{\theta \to 0} \oint \mathbf{A}_+^N d\mathbf{k} = 0 \qquad (m > 0), \tag{21a}$$

$$\lim_{\theta \to \pi} \oint \mathbf{A}_+^N d\mathbf{k} = 2\pi \qquad (m < 0) . \tag{21b}$$

Hence, the choice of the gauge $\mathbf{A}_+^N$ generates a Dirac string with Berry flux of $2\pi$ along $-\hat{\mathbf{e}}_z$, that is the semi-axis $m < 0$. Under a $U(1)$ gauge transformation, wavefunctions in the North gauge transform into wavefunctions in the South gauge with the respective connection $\mathbf{A}^S$ being well defined everywhere but along the $m > 0$ semi-axis. $\mathbf{A}^S$ then generates a Dirac string with a Berry flux of $2\pi$ in the $\hat{\mathbf{e}}_z$ direction, that is along the semi-axis $m > 0$. Generically, under a $U(1)$ gauge transformation, the orientation of the Dirac string is reversed in $(k_x, k_y, m)$ space (see Figs. 2 (a) and (b)).

In both North and South gauges, the Dirac string lies along the $m$ axis. This axis can be rotated in $(k_x, k_y, m)$ space by applying a $SU(2)$ transformation to the Hamiltonian e.g. $\hat{R}_\alpha = \exp\left(-\mathrm{i}\frac{\alpha}{2}\sigma_x\right)$ (Fig. 2 (c)). This amounts to a change of basis of the Hilbert space describing the 2D band crossing in $(k_x, k_y)$ space. Note that under such a unitary transformation, the mass operator is generically modified. Thus, one can distinguish two "gauge choices" that fix the Dirac string: the $SU(2)$ rotations that act on the Hilbert space basis, and that fixes the direction of the Dirac string; and the previously mentioned $U(1)$ gauge transformation on the eigenstates that fixes its orientation.

We illustrate below the pathological situation of the Dirac string lying in the $m = 0$ equatorial plane (Fig. 2 (d)). Let us apply a spin rotation of angle $\pi/2$ around the $x$-axis on the massless 2D Dirac Hamiltonian. The $SU(2)$ spin rotation operator reads

$$\hat{R}_{\pi/2} = \exp\left(-\mathrm{i}\frac{\pi}{4}\sigma_x\right) = \frac{1}{\sqrt{2}}(\mathbb{1} - \mathrm{i}\sigma_x). \tag{22}$$

The Hamiltonian transforms as

$$\hat{R}_{\pi/2} H^{2D} \hat{R}_{\pi/2}^{-1} = k_x \sigma_x + k_y \sigma_z. \tag{23}$$

The rotated Hamiltonian is purely real, and so are the wavefunctions: a possible basis is

$$\psi_+ = \begin{pmatrix} \cos\frac{\varphi}{2} \\ \sin\frac{\varphi}{2} \end{pmatrix} \; ; \; \psi_- = \begin{pmatrix} \sin\frac{\varphi}{2} \\ -\cos\frac{\varphi}{2} \end{pmatrix}, \tag{24}$$

where $\varphi = \arctan k_y/k_x$. The Berry connections $\mathbf{A}_\pm = \mathbf{0}$ are trivial and one could naively conclude that the Berry windings are trivial too, even though the system still has an effective $\mathcal{PT}$ symmetry $\hat{R}_{\pi/2} V \hat{R}_{\pi/2}^{-1}$ (that is proportional to $\mathcal{K}$). However, this contradiction is only apparent. Indeed, let us note that the wavefunctions are $4\pi$ periodic, while $\varphi$ is $2\pi$ periodic. This is a manifestation of a branch cut for the phase $\varphi$, originating from the alignment of the Dirac string with the place $m = 0$, as sketched in figure 2 (d). This branch cut contributes to the integral of the Berry connection along a closed loop circling the origin $\mathbf{k} = \mathbf{0}$, leaving the Berry winding unaffected $\gamma_\pm = \pm\pi$, as it should. Notice that the role of the orientation of the Dirac string and the branch cut has also been pointed out in Ref. [16], where the winding number has been augmented to a *winding vector*.

## 2.5 $n$ band crossings in $D = 2$ and emergent $\mathcal{PT}$

### 2.5.1 Extending the $\mathcal{PT}$ symmetry from $2$ to $n$ band crossings

In the case of graphene, the operator (16) corresponds to the combination of real spatial inversion and time-reversal symmetries, the latter for spinless particles, i.e. where $\sigma_\mu$ does not describe a spin but an orbital degree of freedom. However, in general, it does not need

to be this particular combination : any antiunitary operator $V$ and mass $M$ satisfying the relation (15) will lead to a quantization of Berry winding *via* a $D = 3$ Chern number and emerging $\mathcal{PT}$ symmetry. We now show how this mechanism of quantization of Berry winding can be extended to a generic $n$ band crossing in $D = 2$. The procedure consists of identifying a proper antiunitary transformation $V = U\mathcal{K}$ and mass opening operator $M$ that satisfies the relation (15) in the $D = 3$ extension of the Hamiltonian $H^{3D}(\mathbf{k}, m) = H^{2D}(\mathbf{k}) + m\,M$ Here, the unitary operator $U$ generalizes the $\sigma_x$ matrix and the mass $M$ operator generalizes the $\sigma_z$ matrix from the previous section, where only two-band models were investigated.

We consider a generic linear crossing of $n$ bands described in momentum space $\mathbf{k}$ and in an appropriate basis by the Hamiltonian

$$H^{2D}(\mathbf{k}) = k_x\,\Sigma_1 + k_y\,\Sigma_2, \tag{25}$$

where $\Sigma_1$ and $\Sigma_2$ are $n \times n$ Hermitian matrices that generalize the matrices $\sigma_x$ and $\sigma_y$ of the two-band case. Note that we consider $[\Sigma_1, \Sigma_2] \neq 0$. When this is not the case, both matrices $\Sigma_1$ and $\Sigma_2$ can be diagonalized simultaneously, and the band degeneracy occurs along a nodal line which we do not consider in the present article. We look for an antiunitary operator $V$, which satisfies the relation

$$V^{-1}\Sigma_1 V = \Sigma_1 \; ; \; V^{-1}\Sigma_2 V = \Sigma_2. \tag{26}$$

This operator generalizes that of the 2-band crossing of Eq. (16), and thus plays the role of an effective $\mathcal{PT}$ symmetry, i.e. it is local in $\mathbf{k}$ and antiunitary. The prolongation to $D = 3$ is provided by the following operator

$$M = -\mathrm{i}\,[\Sigma_1, \Sigma_2], \tag{27}$$

which opens a gap at the band crossing and therefore plays a role analogous to $\sigma_z$ for the 2-band crossing. Note that the choice of the prefactor of this operator, and thus the sign of the mass $m$ previously associated to North or South gauges, are arbitrary.

Moreover, this mass operator transforms under the $\mathcal{PT}$ operator as

$$V^{-1}MV = \mathrm{i}[V^{-1}\Sigma_1 V, V^{-1}\Sigma_2 V] = \mathrm{i}[\Sigma_1, \Sigma_2] = -M. \tag{28}$$

Hence the $\mathcal{PT}$ transformation is local in momentum $\mathbf{k}$ and exchanges $m$ and $-m$. This is expressed on the $D = 3$ extension $H^{3D}(\mathbf{k}, m) = H^{2D}(\mathbf{k}) + m\,M$ of the Hamiltonian (25) which satisfies the relation

$$V^{-1}H(\mathbf{k}, m)V = H(\mathbf{k}, -m). \tag{29}$$

This relation generalizes Eq. (15) and leads to a $\pi$ quantization of the Berry winding of a generic $n$ band crossing. Let us now discuss the application of the above procedure in a few specific cases.

### 2.5.2   $\mathcal{PT}$ symmetry for a generic spin band crossing

When the Hamiltonian describes a spin $S$ fermion, up to a unitary spin rotation the Hamiltonian reads

$$H^{2D}(\mathbf{k}) = k_x\,S_x + k_y\,S_y. \tag{30}$$

The spectrum is then a superposition of Dirac-like cones, and a flat band for integer $S$. In this case, the generic mass operator $M$ defined in (27) identifies with $S_z$, such that the natural $D = 3$ extension of the

$$H^{3D}(\mathbf{p}) = k_x S_x + k_y S_y + m S_z = |\mathbf{p}| \, \hat{p}.S_{\hat{p}} \tag{31}$$

where $\hat{p} = \mathbf{p}/|\mathbf{p}|$ of polar coordinates $\theta, \phi$ and $S_{\hat{p}} = \hat{p}.\mathbf{S}$. In the basis of eigenstates of $S_z$, both $S_x = (S_+ + S_-)/2$ and $S_z$ are real while $S_y = (S_+ - S_-)/(2\mathrm{i})$ is purely imaginary, where we used the raising and lowering operators $S_{\pm}$. The action of a $\mathcal{PT}$ transformation $V = U\mathcal{K}$ on $H^{3D}(\mathbf{p})$ translates into

$$V H^{3D}(\mathbf{k}, m) V^{-1} = H^{3D}(\mathbf{k}, -m) \rightarrow U H^{3D}(k_x, k_y, m) U^{-1} = H^{3D}(k_x, -k_y, -m). \tag{32}$$

Thus the unitary transformation $U$ acting on the spin model (31) corresponds to a $\pi$ spin rotation around the $x$ axis

$$U = e^{-\mathrm{i}\pi S_x}. \tag{33}$$

For the particular case of spins $S = \frac{1}{2}$, using $S_x = \sigma_x/2$ we recover the transformation (16) with $U = \sigma_x$.

### 2.5.3  Real and imaginary Hamiltonians

Of particular interest is the case of a real Hamiltonian, corresponding to two purely real Hermitian generators $\Sigma_1$ and $\Sigma_2$. This generalizes the situation represented in Fig. 2 (d) : while the eigenstates can be chosen real and the Berry connexion vanishes, the Berry winding is still finite and is solely determined by a discontinuity of the eigenstates around the band crossing point which is a manifestation of the Dirac string located in the plane $m = 0$. In this situation the mass operator $M$ in (27) is purely imaginary, and the $\mathcal{PT}$ operator is simply given by the complex conjugation $V = \mathcal{K}$, *i.e.* $U = \mathbb{I}$. Notice that this argument remains valid also in the case where $\Sigma_1$ and $\Sigma_2$ are not purely real but are related to real operators by a global unitary transformation $U$, $U^{-1}\Sigma_{1/2}U$, in which case $V = U\mathcal{K}$.

The opposite case of a purely imaginary Hamiltonian, corresponding to imaginary Hermitian matrices $\Sigma_1$ and $\Sigma_2$ is also of interest. More precisely, we consider $\Sigma_1 = \Sigma_y^{m_1, m_1'}$, $\Sigma_2 = \Sigma_y^{m_2, m_2'}$ matrices, where $\Sigma_y^{m, m'}$ generalizes the $\sigma_y$ Pauli matrix and has only two non-zero elements in line $m$ and column $m'$ (and line $m'$ and column $m$, see appendix B). Their commutator vanishes unless both matrices share at least a common line of non-zero entries. Let us consider without loss of generality that the common line of non-zero entries is $m_1 = m_2$. In this case, the corresponding mass matrix $M$ (27) is also a $\Sigma_y$ matrix, $M = i\Sigma_y^{m_1', m_2'}$. These three matrices form an SU(2) subalgebra embedded in SU($n$), as one may easily show with the help of the asymmetry (63) and the commutation relations (62) of the $\Sigma$-matrices [see Eqs. (63) and (62) of Appendix B] Most saliently, one may construct explicitly a $\mathcal{PT}$ symmetry operator $V = U\mathcal{K}$ whose unitary SU($n$) matrix is simply given by the diagonal matrix $U = \mathrm{Diag}(1 - 2\delta_{m, m_1})$ which satisfies the relations (26), see Appendix B.

## 3  Berry winding in three-band models *via* examples

We have seen with the simple example of the two-band crossing how a general topological characterization of a 2D band crossing can be obtained. Considering a gap opening term,

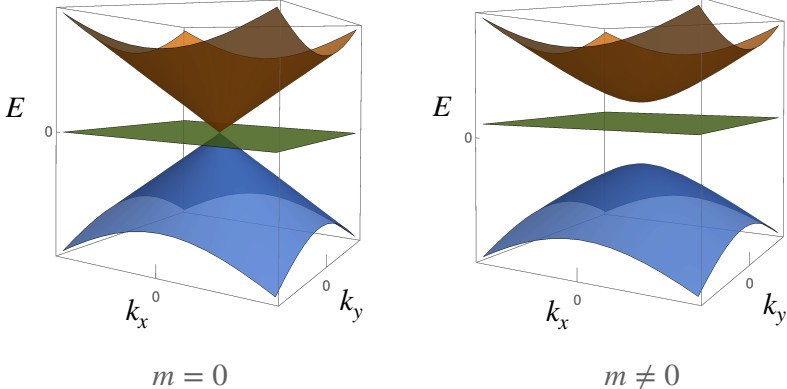

Figure 3: Three-fold band crossing point with a linear dispersion relation occurring in the band structure of the Lieb, $H_3$ and $T_3$ models. A non zero mass term $m \neq 0$ lifts the degeneracy in all models.

we consider the 2D crossing as a critical phase separating gapped phases and characterize the 2D quantized Berry winding as a 2D projection of a 3D Chern number. This projection is associated to the presence of an antiunitary transformation between the "northern" and "southern" sides of the 2D plane containing the crossing, see Fig. 1. Let us now follow our investigation by considering 2D semi-metallic phases beyond two band crossings, the simplest examples of which being a three-band crossing. Following the reasoning of the previous section, we study local Hamiltonians describing the crossings and investigate whether they can be characterized by a quantized Berry winding that is obtained from the 2D projection of a 3D Chern number.

The 2D three-band crossings we consider correspond to the modified dice lattice, so-called $\alpha$-$T_3$ model [17], the three-band hexagonal, so-called $H_3$ model [18] and the Lieb lattice model [19]. The motivation for this choice is the following: all three models have the same spectrum shown in Fig. 3 but have fundamentally different topological properties as we show below. While the $H_3$ model has a hidden spin $S = 1/2$ SU(2) symmetry and is thus characterized by the same topological properties as our simple two-band Hamiltonian (1), the Lieb lattice yields a low-energy spin $S = 1$ SU(2)-symmetric model. Both models therefore have a $k$-local $\mathcal{PT}$ symmetry. This needs to be contrasted with the $\alpha$-$T_3$ model, which interpolates in the low-energy limit between the $H_3$ and the Lieb models and that has generally no emergent $\mathcal{PT}$ symmetry, apart from the parameter sets that correspond precisely to the $H_3$ and Lieb models.

## 3.1   The hexagonal three-band $H_3$ model

First we consider a 3-band extension of the tight-binding model of graphene, simply obtained by adding a second atomic orbital on one of the two sublattices of the hexagonal lattice, see Fig. 4c. This $H_3$ model possesses 3-band crossings at points $\mathbf{K}$ and $\mathbf{K}'$ of the 2D Brillouin Zone. The addition of an extra orbital on only one of the two sublattices breaks various crystalline symmetries of the honeycomb lattice. In particular, both parity $\mathcal{P}$ and intravalley mirror symmetry are lost. Yet, we show that an $\mathcal{PT}$ symmetry emerges at low energy, leading to quantized Berry windings. This emphasizes the role of emergent symmetries that are *local*

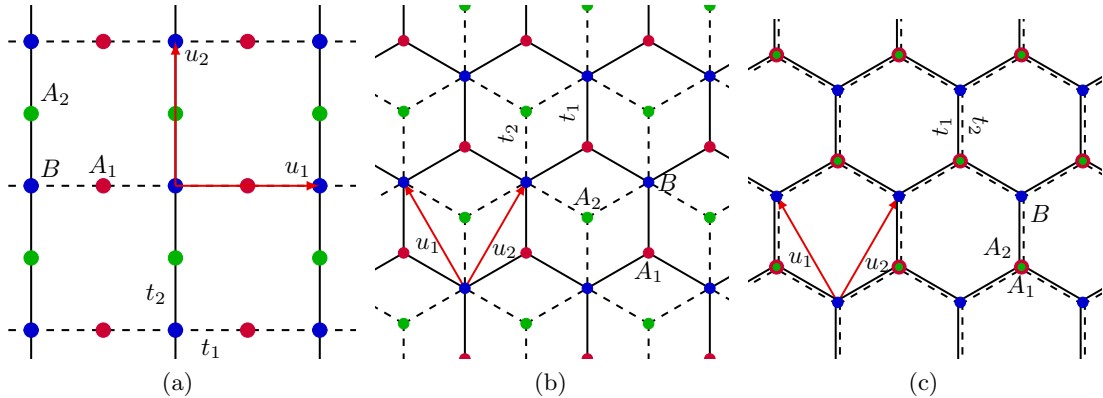

Figure 4: Schematic representation of the tight-binding models : (a) The Lieb lattice model, (b) The $\alpha$-T$_3$ model, (c) The three band hexagonal (H$_3$) model, corresponding to a honeycomb lattice with two decoupled states on one sublattice. In all three cases bonds between sites correspond to nearest neighbour hoppings, between $A_i$ and $B$ only. The $\mathbf{u}_i$ are Bravais lattice unit vectors.

*in momentum* in the quantization of Berry windings. Furthermore, we relate the topological properties of the H$_3$ model to those of an effective $S = 1/2$ spin of 2 bands with a spectator band. Indeed, the $\Sigma_1$ and $\Sigma_2$ matrices describing the crossing complemented by a mass operator $M$ form a $SU(2)$-like subgroup of $SU(3)$. In a nutshell, we show that the H$_3$ model has the unique feature of having none of the crystal symmetries of graphene but the same $\mathbf{k}$-local topological properties.

Around the $\mathbf{K}$ crossing point, the low-energy Bloch Hamiltonian describing the band crossing of the H$_3$ model takes the form

$$H_{2D}^{H_3}(\mathbf{k}) = \begin{pmatrix} 0 & 0 & \cos\beta \ (k_x - \mathrm{i}k_y) \\ 0 & 0 & \sin\beta \ (k_x - \mathrm{i}k_y) \\ \cos\beta \ (k_x + \mathrm{i}k_y) & \sin\beta \ (k_x + \mathrm{i}k_y) & 0 \end{pmatrix} , \qquad (34)$$

in the $(A_1, B, A_2)$ basis, with $\tan\beta = t_1/t_2$ where $t_1$ and $t_2$ are the two amplitudes of nearest neighbor couplings, see Fig. 4c. The spectrum consists in two linearly dispersing bands $\varepsilon_\pm = \pm k$ and a flat band $\varepsilon_0 = 0$, as shown in Fig. 3. The eigenstates of this H$_3$ model are characterized by quantized Berry windings similar to graphene : $\gamma_\pm = -\pi$ and $\gamma_0 = 0$. This quantization is enforced by the presence of a $\mathcal{PT}$ symmetry of the model, in spite of the lack of parity. This emergent $\mathcal{PT}$ symmetry does not hold at the microscopic level, but only in the low energy limit around the band crossing point. It corresponds to the operator :

$$V = \begin{pmatrix} \sin^2\beta & -\cos\beta \ \sin\beta & \cos\beta \\ -\cos\beta \ \sin\beta & \cos^2\beta & \sin\beta \\ \cos\beta & \sin\beta & 0 \end{pmatrix} \mathcal{K}. \qquad (35)$$

This symmetry can be inferred from the representation of the low-energy H$_3$ Hamiltonian

model as a 2-band crossing Dirac Hamiltonian with a disconnected flat band

$$H'(\mathbf{k}) = W^{-1}H(\mathbf{k})W = \begin{pmatrix} 0 & 0 & 0 \\ 0 & 0 & k_- \\ 0 & k_+ & 0 \end{pmatrix} \text{ with } W = \begin{pmatrix} -\sin\beta & \cos\beta & 0 \\ \cos\beta & \sin\beta & 0 \\ 0 & 0 & 1 \end{pmatrix}. \tag{36}$$

and $k_\pm = k_x \pm ik_y$. In this representation, the $\mathcal{PT}$ symmetry is a simple transformation

$$V' = U_2^{-1}VU_2 = \begin{pmatrix} 1 & 0 & 0 \\ 0 & 0 & 1 \\ 0 & 1 & 0 \end{pmatrix}\mathcal{K}. \tag{37}$$

Let us now show how this $\mathcal{PT}$ symmetry relates the quantization of the Berry windings to the Chern number of bands in a $D = 3$ dimensional extension of the model. For this purpose, we choose the following mass operator

$$M = -\mathrm{i}[\Sigma_1, \Sigma_2] = 2 \begin{pmatrix} \cos^2\beta & \cos\beta\,\sin\beta & 0 \\ \cos\beta\,\sin\beta & \sin^2\beta & 0 \\ 0 & 0 & -1 \end{pmatrix},$$

corresponding, in the representation (36), to a standard $\sigma_z$ mass operator :

$$M' = W^{-1}MW = 2 \begin{pmatrix} 0 & 0 & 0 \\ 0 & 1 & 0 \\ 0 & 0 & -1 \end{pmatrix}. \tag{38}$$

Hence, in the representation (36), the relevant operators in the effective two-level systems identifies with those discussed in section 2 :

$$H', \Sigma'_{1/2}, M', V'\mathcal{K} \sim \begin{pmatrix} c_1 & 0 \\ 0 & c_2\sigma_\mu \end{pmatrix}, \tag{39}$$

where $\sigma_\mu$ are the Pauli matrices and $c_1$ and $c_2$ are scalars[2]. Therefore the $H_3$ model has an underlying SU(2) symmetry that allows us to relate its topological properties to those of two-band models such as graphene.

The spectrum of the model $H_{3D}^{H_3}(\mathbf{k}, m) = H_{2D}^{H_3}(\mathbf{k}) + mM$ reads $\varepsilon_0 = 0$, $\varepsilon_\pm = \pm\sqrt{k^2 + 4m^2}$. The eigenstates of this model are conveniently discussed by introducing the the vector $\mathbf{p} = (k_x, k_y, 2m)$ in spherical coordinates $(p, \theta, \varphi)$, see appendix C. Let us focus on the eigenstates of the band $\varepsilon_+$ band. For this band, we choose two $N$ and $S$ gauges smoothly defined respectively for positive and negative masses $m$. The associated Berry connections are, see appendix C :

$$\mathbf{A}_+^N = -\sin^2\frac{\theta}{2}\boldsymbol{\nabla}_\mathbf{p}\varphi \; ; \; \mathbf{A}_+^S = \cos^2\frac{\theta}{2}\boldsymbol{\nabla}_\mathbf{p}\varphi. \tag{40}$$

The Berry connection in the northern gauge describes a Berry monopole at $\mathbf{p} = \mathbf{0}$ together with a Dirac string of flux $2\pi\hat{\mathbf{e}}_z$ along the $\mathbf{k} = \mathbf{0}, m < 0$ semi-axis. The same monopole with the symmetric string (along the $\mathbf{k} = \mathbf{0}, m > 0$ semi-axis) is described by the Berry connection

---

[2]If the constant $c_1$ were non-zero in the Hamiltonian, it would allow us to shift the energy of the decoupled single band and provide it with an additional dispersion.

in the southern gauge. On the equator, both connections are smoothly defined and related by a gauge transformation $\mathbf{A}_+^N = \mathbf{A}_+^S - \boldsymbol{\nabla}_{\mathbf{k}}\varphi$ leading to a Chern number

$$\nu_+ = \frac{1}{2\pi} \oint_{\partial \mathcal{S}_N} \left( \mathbf{A}_+^N - \mathbf{A}_+^S \right) \cdot d\boldsymbol{\ell} = -1 \; ; \; \gamma_+ = \pi\nu_+. \tag{41}$$

The presence of the emergent $\mathcal{PT}$ symmetry (35) enforces a quantization of the 2D Berry winding $\gamma_+ = \pi\nu_+$ deduced from the Chern number. Besides the above procedure allows one to reveal the underlying spin-1/2 structure and topological properties of the H$_3$ model, identical to that of graphene, in spite of the difference in crystalline symmetries and parity between both models. This stresses the importance of an effective emergent $\mathcal{PT}$ symmetry in the Berry winding quantization even though symmetries $\mathcal{P}$ or $\mathcal{T}$ are absent at the microscopic level.

## 3.2   Spin-1 and microscopic $\mathcal{PT}$ symmetry on the Lieb lattice

The Lieb lattice model [19] is a bipartite lattice model on a square lattice, with three sites per unit cell, as shown on Fig. 4a. The three bands cross linearly at $\mathbf{k} = \mathbf{0}$ in the Brillouin Zone, and the low-energy Hamiltonian near the crossing takes the form

$$H_{2D}^{\text{Lieb}}(\mathbf{k}) = \begin{pmatrix} 0 & 0 & -ik_x \\ 0 & 0 & ik_y \\ ik_x & -ik_y & 0 \end{pmatrix} = k_x\,\Sigma_1 + k_y\,\Sigma_2 \tag{42}$$

with a spectrum $\varepsilon_0 = 0, \varepsilon_\pm = \pm k$ shown in Fig. 3. Eigenstates of the Lieb model possess an effective spin-1. In the following, we apply the general procedure of section 2.5 for such a local spin-1 band structure. The mass operator $M$ together with the $\Sigma_i$ matrices indeed form a spin algebra. Non trivial Chern numbers $\nu = 0, \pm 2$, distinct from those of the previous spin-1/2 case, are identified for the 3 bands with a finite mass. The presence of a local $\mathcal{PT}$ symmetry leads to a topological quantization of the Berry winding $2\pi$ around the band crossing.

Similarly to graphene, the nearest neighbor tight-binding model on the 2D Lieb lattice is invariant under both parity $\mathcal{P}$ and time-reversal symmetry $\mathcal{T}$. The combination of both is thus also a symmetry of the 2D Hamiltonian and corresponds to the required operator $V$:

$$V = \begin{pmatrix} 1 & 0 & 0 \\ 0 & 1 & 0 \\ 0 & 0 & -1 \end{pmatrix} \mathcal{K}. \tag{43}$$

Following the lines of reasoning of section (2.5), we define a mass operator $M = -i[\Sigma_1, \Sigma_2]$ leading to the 3D extension of (42) :

$$H_{3D}^{\text{Lieb}}(\mathbf{k}, m) = k_x\Sigma_1 + k_y\Sigma_2 + mM = \begin{pmatrix} 0 & im & -ik_x \\ -im & 0 & ik_y \\ ik_x & -ik_y & 0 \end{pmatrix}. \tag{44}$$

The spectrum is $\varepsilon_0 = 0, \varepsilon_\pm = \pm\sqrt{k^2 + m^2}$. In this case, the matrices $\Sigma_1, \Sigma_2$ and $M = \Sigma_3$ are a purely imaginary representation of the spin $S = 1$ algebra,

$$[\Sigma_\mu, \Sigma_\nu] = i\epsilon_{\mu\nu\sigma}\Sigma_\sigma, \tag{45}$$

in terms of the totally antisymmetric tensor $\epsilon_{\mu\nu\sigma}$. Besides the spin structure, the matrices $\Sigma_1, \Sigma_2, M$ correspond exactly to the case of purely imaginary matrices discussed in section 2.5.

Therefore, we illustrate with the Lieb-lattice model the general procedure discussed in this section 2.5. Similarly to the $H_3$ model, we obtain a low-energy SU(2) symmetry, as it is shown by the commutation relations (45). However, in the present case, we are confronted with a purely imaginary representation of SU(2), which requires being embedded in a higher-dimensional space (here in terms of $3 \times 3$ matrices) and thus a larger spin (here $S = 1$).

First we consider two smooth gauge choices for eigenstates around the band crossing, corresponding to a covering of the sphere in $\mathbf{p} = (k_x, k_y, m)$ space by two hemispheres $\mathcal{S}_N$ and $\mathcal{S}_S$. Using polar coordinates, we obtain that away from the $m$-axis the two conventions are related by the unitary transformation $\psi_\pm^N = e^{\pm 2i\varphi} \psi_\pm^S$ and the Berry connections read

$$\mathbf{A}_+^N = (1 - \cos\theta)\boldsymbol{\nabla}_{\mathbf{p}}\varphi \; ; \; \mathbf{A}_+^S = -(1 + \cos\theta)\boldsymbol{\nabla}_{\mathbf{p}}\varphi. \tag{46}$$

The obstruction to define a smooth gauge everywhere manifests itself through a Dirac string located along the negative $m$ axis for the $N$ gauge, and carrying a flux $4\pi\hat{\mathbf{e}}_z$, as illustrated in Fig. 2 (see also Appendix D). The presence of this half tube of flux can be detected by computing the Berry flux threading a disk of radius $k\sin\theta$ centered on the $m$ axis in the limit of a vanishing polar angle $\theta$.

The presence of the (real) $\mathcal{PT}$ symmetry implies that on the equator where $m = 0$ (or $\theta = \pi/2$), the Berry connections verify $\mathbf{A}_\pm^N = -\mathbf{A}_\pm^S = \pm\boldsymbol{\nabla}_{\mathbf{k}}\varphi$ leading to a quantization relation of the 2D Berry winding

$$\gamma_\pm^N = \oint_{m=0} \mathbf{A}_\pm^N d\mathbf{k} = \frac{1}{2}\oint_{m=0} \left(\mathbf{A}_\pm^N - \mathbf{A}_\pm^S\right) d\mathbf{k} = \pi\nu_\pm \tag{47}$$

with the 3D Chern number

$$\nu_\pm = \frac{1}{2\pi}\oint \left(\mathbf{A}_\pm^N - \mathbf{A}_\pm^S\right) d\mathbf{k} = \pm 2, \tag{48}$$

see appendix D for a detailed derivation. Note that here a Berry winding of $2\pi$ is distinct from a Berry winding of 0, in contrast with a Wilson loop characterization (see the discussion in Appendix A).

Thus, we have illustrated our procedure on a particular model that displays 3-band crossing point with a spin-1 structure emerging from the linearization of the Lieb lattice.

## 3.3 A model without any effective $\mathcal{PT}$ symmetry: the $\alpha$-$\mathbf{T}_3$ model

Let us now study a model that interpolates between a spin-1 and a spin-1/2 band structure, the $\alpha$-$T_3$ model. This model is a natural extension of graphene which consists of adding an extra atomic site located in the centre of each hexagon of the honeycomb lattice, as shown in Fig. 4b. This extra site is coupled *via* nearest neighbor coupling of amplitude $t_2$ to only one ($B$) of the two sublattices of the honeycomb lattice, and the coupling between nearest neighbor sites of the honeycomb lattice has an amplitude $t_1$. This bipartite structure preserves the chiral symmetry and constrains the spectrum to be symmetric around $E = 0$. The 3 sites per unit cell lead to a flat band at $\epsilon_0 = 0$ with two finite energy bands $\epsilon_+(\mathbf{k}) = -\epsilon_-(\mathbf{k})$ which cross linearly at the $\mathbf{K}$ and $\mathbf{K}'$ points of the honeycomb lattice's Brillouin Zone. Around the $\mathbf{K}$ point, the crossing is described by the Hamiltonian

$$H_{2D}^{T_3}(\mathbf{k}) = k_x\, \Sigma_1 + k_y\, \Sigma_2 \tag{49}$$

$$= \begin{pmatrix} 0 & 0 & \cos\beta\,(k_x - ik_y) \\ 0 & 0 & \sin\beta\,(k_x + ik_y) \\ \cos\beta\,(k_x + ik_y) & \sin\beta\,(k_x - ik_y) & 0 \end{pmatrix}, \tag{50}$$

where $\tan\beta = t_1/t_2$ gives the relative strength of nearest-neighbour hoppings, and is usually denoted by $\alpha$. When $\beta = 0$ or $\pm\pi/2$, one of the $A_i$ sublattices becomes disconnected from the rest of the lattice. In this particular case the model corresponds to an effective spin-1/2 with a spectator flat band, *i.e.* we retrieve the $H_3$ model in this limit. In the symmetric case $\beta = \pm\pi/4$, *i.e.* when $t_1 = \pm t_2$, the matrices $\Sigma_1$, $\Sigma_2$ and $M = -\mathrm{i}[\Sigma_1, \Sigma_2]$ form a spin-1 algebra, and this limiting case the model identifies with the Lieb model. Therefore the $\alpha$-$T_3$ model can be interpreted as a smooth interpolation between $S = 1/2$ and $S = 1$ structures.

In this model, the Berry windings of the different bands around the band crossing point are not quantized and vary continuously with the parameter $\beta$ [17] :

$$\gamma_\pm = -\pi \cos 2\beta, \qquad \gamma_0 = 2\pi \cos 2\beta. \tag{51}$$

Let us now relate these values of the Berry winding to the mechanism of quantization discussed in Sec. 2.5. In the $\alpha$-$T_3$ model, inversion symmetry is broken when the hoppings are unequal $t_1 \neq t_2$, *i.e.* for $\beta \neq \pi/4$. Hence the lattice of the $\alpha$-$T_3$ model does not possess a microscopic $\mathcal{PT}$ symmetry. Besides, we show below that its Hamiltonian (51) around a band crossing lacks any emergent $\mathcal{PT}$ symmetry. Indeed, the unitary part $U$ of such an antiunitary transformation $V = U\mathcal{K}$ must satisfy

$$\Sigma_1 U = \Sigma_1 U, \qquad \Sigma_2 U = -\Sigma_2 U. \tag{52}$$

Solving this linear algebra equations, we find solutions only for the above symmetric cases:

$$U(\beta = 0) = \begin{pmatrix} 0 & 0 & 1 \\ 0 & \lambda & 0 \\ 1 & 0 & 0 \end{pmatrix} ; U(\beta = \pm\frac{\pi}{4}) = \begin{pmatrix} 0 & 1 & 0 \\ 1 & 0 & 0 \\ 0 & 0 & 1 \end{pmatrix} ; U(\beta = \pm\frac{\pi}{2}) = \begin{pmatrix} \lambda & 0 & 0 \\ 0 & 0 & 1 \\ 0 & 1 & 0 \end{pmatrix}. \tag{53}$$

For $\beta \neq 0, \pm\pi/4, \pm\pi/2$, no effective $\mathcal{PT}$ symmetry exists, and the scenario of Sec.2.5 of topological quantization of the Berry windings does not hold, in agreement with the values (51) which are integer multiple of $\pi$ only for $\beta = 0, \pm\pi/4, \pm\pi/2$.

Note that, however, the absence of the effective $\mathcal{PT}$ does not prevent the Chern numbers of the extended 3D model to be non zero. To illustrate this point, we consider a simpler mass term than in (27), allowing for a simple analytical analysis of the Chern numnber, corresponding to the 3D extension of the $\alpha$-$T_3$ model as

$$H_{3D}^{T_3}(\mathbf{k}, m) = H_{2D}^{T_3}(\mathbf{k}) + mM \tag{54}$$

$$= \begin{pmatrix} m & 0 & \cos\beta \; (k_x - \mathrm{i}k_y) \\ 0 & -m & \sin\beta \; (k_x + \mathrm{i}k_y) \\ \cos\beta \; (k_x + \mathrm{i}k_y) & \sin\beta \; (k_x - \mathrm{i}k_y) & -\cos 2\beta \; m \end{pmatrix}, \tag{55}$$

The spectrum displays a flat band $\varepsilon_0 = -m\cos 2\beta$ and two linearly dispersive bands $\varepsilon_\pm = \pm\sqrt{k^2 + m^2}$. The eigenstates of the positive energy band $\epsilon_+$ are associated with a Chern number $\nu_+ = 2$ for any surface enclosing the band crossing point, and irrespective of the value of $\beta$. This can be calculated by considering $N$ and $S$ gauge choices, valid respectively for $m > 0$ and $m < 0$: wavefunctions in the two gauges are related through the transformation $\psi_+^N = \exp(2i\varphi)\psi_+^S$ in polar coordinate, leading to the relation between the corresponding Berry connections $\mathbf{A}_+^N = \mathbf{A}_+^S + 2\boldsymbol{\nabla}_\mathbf{k}\varphi$. Detailed expression of the connections can be found in appendix E. The Chern number is deduced from the relation

$$\nu_+ = \frac{1}{2\pi} \oint_{m=0} (\mathbf{A}_+^N - \mathbf{A}_+^S) \, d\mathbf{k} = \frac{1}{2\pi} \oint_{m=0} 2\boldsymbol{\nabla}_\mathbf{k}\varphi \, d\mathbf{k} = 2. \tag{56}$$

For $\beta = \pi/4$, a spin-1 algebra is recovered corresponding to a $\nu = 2$ Chern number for the $\epsilon_+$ band. The Chern number of the gapped bands being topological properties of these bands, it remains unchanged as $\beta$ varies away from $\pi/4$ given than no gap closes. When $\beta = 0$ or $\pi/2$, a topological transition occurs : a gap closes with the flat band touching one of the dispersive bands and the Hamiltonian then describes an effective spin-1/2 structure.

As in the 2-band crossing case, this non zero Chern number manifests an obstruction to smoothly define eigenstates in $(\mathbf{k}, m)$ space. This leads to the presence of a Dirac string or half flux tube originating from the Berry monopole for any choice of gauge (see Fig. 2). We show the existence of this Dirac string by calculating for *e.g.* the N gauge valid for $m > 0$ the flux threading a disk of radius $k \sin\theta$ centered on the $m$ axis, in the limit $\theta \to 0, \pi$, see Appendix E :

$$\lim_{\theta \to 0} \oint \mathbf{A}_+^{\mathbf{N}} d\mathbf{k} = 0, \; ; \; \lim_{\theta \to \pi} \oint \mathbf{A}_+^{\mathbf{N}} d\mathbf{k} = \frac{8\pi \sin^2 \beta}{1 - \cos 2\beta}. \tag{57}$$

As opposed to the previous models, for $\beta \neq 0, \pm\pi/4, \pm\pi/2$ the flux is not quantized in units of $2\pi$. This non-quantization reflects the non quantization of the 2D Berry windings (51).

The above results illustrate that a given model, here the $\alpha$-T$_3$ model, can possess non-vanishing Chern numbers when a gap is opened, but unquantized 2D Berry windings if no emergent $\mathcal{PT}$ symmetry is present at low energy. This corresponds to a situation where the singularity line of each gauge choice, generalizing the Dirac strings, are associated with unquantized and gauge dependent Berry fluxes.

The above discussion of the $\alpha$-T$_3$ model extends to the critical HgCdTe material. For a critical Cd concentration, a linear crossing occurs between three doubly degenerate bands in Hg$_{1-x}$Cd$_x$Te [20]. This critical semi-metallic phase provides a 3D extension of the $\alpha$-T$_3$ model for a specific value of the parameter $\tan\beta = \alpha = \frac{1}{\sqrt{3}}$ [21]. Although the 3D phase is trivial with Chern number $\nu = 0$, it projects onto a 2D crossing with non zero, although non quantized, Berry windings, see Appendix E.2.

# 4 Conclusions and perspectives

In this article, we have discussed a necessary condition for the Berry winding of eigenstates around a $D = 2$ band crossing to be quantized. This condition is based on the existence of a $k$-local $\mathcal{PT}$ symmetry around the band crossing, which allows the Berry winding to inherit a topological robustness from the Chern number of the bands when a gap is opened. As a consequence, the topological nature of the quantized Berry winding encodes a robustness of the eigenstates with respect to $\mathcal{PT}$ preserving perturbations. We have illustrated this interplay between a $\mathcal{PT}$ symmetry, quantized Berry windings and Chern number on several 3-band crossing occuring in $D = 2$ lattice models, the H$_3$, Lieb lattice and $\alpha$-T$_3$ models. While the SU(2)-structure of H$_3$ and the Lieb models, for spins $S = 1/2$ and $S = 1$, respectively, in the low-energy limit allows for the emergence of a $k$-local $\mathcal{PT}$ symmetry, this is generally not the case in the $\alpha$-T$_3$ model. Indeed, the latter interpolates between the H$_3$ and the Lieb models, and no such symmetry exists except at the two limits. In Ref. [18] the existence of a quantized Berry winding is related to a non vanishing minimal conductivity at the crossing, which originates from the nature of evanescent states in a finite geometry. It is thus tempting to speculate that the nature of evanescent states associated to a band crossing depends on the presence of a $\mathcal{PT}$, a question worth exploring in future work.

As pointed out in the main text, the connection between 2D winding number and the presence of a quantized monopole in the 3D embedding space is associated with a Dirac string along which the Berry connection is not defined. The orientation of the Dirac string and thus the definition of the $k$-local $\mathcal{PT}$ symmetry are nevertheless gauged dependent. Future studies may involve the evolution of this approach beyond strict $k$-locality, e.g. in the vicinity of a merging point where two band-contact points unite. It has been shown that the nature of such a merging transition depends on the winding number of the Dirac points [16, 22–24]. It would be interesting to check whether the different merging transitions could be classified within a $\mathcal{PT}$ symmetry that could be defined in the neighbourhood of the merging point in $k$-space that contains both band-contact points. Such future study would thus deal with "second-generation continuum models" beyond the linear-band approximation [25] and patches in reciprocal space. This is, however, beyond the scope of the present paper which discusses a strictly $k$-local $\mathcal{PT}$ symmetry.

## Acknowledgements

The authors would like to thank Frédéric Piéchon for fruitful discussion.

**Funding information** M.O.G and D.C. would like to acknowledge financial support from Agence Nationale de la Recherche (ANR project "Dirac3D") under Grant No. ANR-17-CE30-0023. D.C. and P.D. acknowledge financial support from the IDEXLYON breakthrough program ToRe.

## A Berry windings versus Wilson loop flow

The Wilson loop operator is the path ordered exponential of the integral of the Berry connexion $\mathbf{A} = -\mathrm{i}\langle\psi|\boldsymbol{\nabla}_{\mathbf{k}}|\psi\rangle$ along a loop

$$W[\ell] = \overline{\exp}\left(-\mathrm{i}\oint_\ell \mathbf{A}(\mathbf{k})d\mathbf{k}\right). \tag{58}$$

When the loop is non contractible, it corresponds to a Zak phase [26]. The Berry winding along loop $\ell$ is by definition

$$\gamma_\ell = \oint_\ell \mathbf{A}(\mathbf{k})d\mathbf{k}. \tag{59}$$

The relation between Berry winding and Wilson loop is thus

$$\log W[\ell] = \gamma_\ell \bmod 2\pi, \tag{60}$$

depending on the choice for the determination of the complex logarithm.

In the case of the $3D$ extension of the Lieb lattice, we have seen that $\gamma = 0 \neq \gamma = 2\pi$ for the Berry windings. In this model, a $2\pi$ Berry winding is related to a Chern number $|\nu| = 2$. With this result we emphasize that Berry windings and Wilson loops encode different topological properties, relative to different classes of perturbations.

# B   $\mathcal{PT}$ symmetry for an imaginary Hamiltonian

We focus on the situation discussed in section 2.5 where $\Sigma_1, \Sigma_2$ and $M$ matrices are purely imaginary. From $n \geq 3$ on, the basis of Hermitian $n \times n$ matrices used to construct an $n$-band Bloch Hamiltonian contains at least three, indeed $n(n-1)/2$ purely imaginary matrices, and the commutation prescription to generate a mass term yields an SU(2) subgroup embedded in SU($n$).

Let us consider an $n \times n$ generalization of Pauli $\sigma_y$ matrices, where the subset of purely imaginary matrices contains $\Sigma_1$ and $\Sigma_2$, which have only two non-zero elements in line $m_{1/2}$ and column $m'_{1/2}$, and naturally its complex conjugate in line $m'_{1/2}$ and column $m_{1/2}$. Their commutator vanishes unless both matrices share at least a common row of non-zero entries. In order to see this point, let us consider two imaginary Hermitian matrices $\Sigma^{m_1,m'_1}$ and $\Sigma^{m_2,m'_2}$, where the notation indicates that all entries are zero apart from the element in line $m_{1/2}$ and column $m'_{1/2}$, which is $i$, and that in line $m'_{1/2}$ and column $m_{1/2}$, which is $-i$. In components, these matrices can be generically written as

$$\Sigma^{m_0,m'_0}_{m,m'} = i \left( \delta_{m,m_0} \delta_{m',m'_0} - \delta_{m,m'_0} \delta_{m',m_0} \right). \tag{61}$$

The components of the commutator (times the imaginary $i$ in order to obtain a purely imaginary Hermitian operator) are readily calculated and read

$$\begin{aligned} i \left[ \Sigma^{m_1,m'_1}, \Sigma^{m_2,m'_2} \right]_{m,m'} &= i \Big[ \delta_{m_1,m_2} (\delta_{m,m'_1} \delta_{m',m'_2} - \delta_{m,m'_2} \delta_{m',m_1}) \\ &\quad + \delta_{m'_1,m'_2} (\delta_{m,m_1} \delta_{m',m_2} - \delta_{m,m_2} \delta_{m',m_1}) \\ &\quad - \delta_{m_1,m'_2} (\delta_{m,m'_1} \delta_{m',m_2} - \delta_{m,m_2} \delta_{m',m'_1}) \\ &\quad - \delta_{m'_1,m_2} (\delta_{m,m_1} \delta_{m',m'_2} - \delta_{m,m'_2} \delta_{m',m_1}) \Big]. \end{aligned} \tag{62}$$

Notice first that the commutator only gives a non-zero operator when the original matrices $\Sigma^{m_1,m'_1}$ and $\Sigma^{m_2,m'_2}$ share at least a common row of non-zero entries. Furthermore, there is a redundancy in the description because by definition

$$\Sigma^{m_0,m'_0} = -\Sigma^{m'_0,m_0} \tag{63}$$

so that one can omit the last two lines with a negative sign in Eq. (62). Let us consider without loss of generality that the common line of non-zero entries is $m_1 = m_2$. The commutator then yields another matrix $\Sigma^{m'_1,m'_2}$,

$$\left[ \Sigma^{m'_1,m_1}, \Sigma^{m_1,m'_2} \right] = i \Sigma^{m'_1,m'_2} = iM, \tag{64}$$

which is nothing other than the mass operator, by construction. Moreover, one notices that these three matrices form an SU(2) subalgebra embedded in SU($n$), as one may easily show with the help of the asymmetry (63) and the commutation relations (62) of the $\Sigma$-matrices.

Most saliently, one may also construct explicitly a $\mathcal{PT}$ symmetry operator $V = U\mathcal{K}$ whose unitary SU($n$) matrix is simply given by the diagonal matrix

$$U_{m,m'} = \delta_{m,m'}(1 - 2\delta_{m,m_1}), \tag{65}$$

of elements 1 apart from the $m_1$-th line and column, where the element is $-1$. For $m_1 \neq m_1'$, case that is excluded because we consider Hermitian matrices, one has

$$(U\Sigma^{m_1,m_1'})_{m,m'} = -i(\delta_{m,m_1}\delta_{m',m_1'} + \delta_{m,m_1'}\delta_{m',m_1}) = -(\Sigma^{m_1,m_1'}U)_{m,m'}, \tag{66}$$

that means that $U$ anticommutes with both $\Sigma^{m_1,m_1'}$ and $\Sigma^{m_1,m_2'}$, and since the latter are purely imaginary, we have

$$V^{-1}\Sigma^{m_1,m'}V = \Sigma^{m_1,m}, \tag{67}$$

as required by the $\mathcal{PT}$ symmetry. Furthermore, $U$ commutes with the mass operator $\Sigma^{m_1',m_2'}$ because the lines $m_1' \neq m_1$ and $m_2' \neq m_1$ remain invariant by multiplication with $U$, which only acts as the one-matrix here. Consequently, $V$ anticommutes with the mass operator, as required.

## C  Topological properties of the $H_3$ model

The 3D $H_3$ Hamiltonian reads:

$$H = k_x\Sigma_1 + k_y\Sigma_2 + mM = \begin{pmatrix} 2m\cos^2\beta & 2m\sin\beta\cos\beta & \cos\beta k_- \\ 2m\cos\beta\sin\beta & 2m\sin^2\beta & \sin\beta k_- \\ \cos\beta k_+ & \sin\beta k_+ & -2m \end{pmatrix}. \tag{68}$$

Its spectrum is $\varepsilon_0 = 0, \varepsilon_\pm = \pm\sqrt{k_x^2 + k_y^2 + 4m^2}$. Note that the velocity along the $m$ axis differs by a factor 2 from the Lieb model. This is due to the difference of commutation relations between spin-1 operators for the Lieb model and spin-1/2 operators for the $H_3$ model. Let us introduce the spherical coordinates $\mathbf{p} = (k_x, k_y, 2m) = p(\sin\theta\,\cos\varphi, \sin\theta\,\sin\varphi, \cos\theta)$. We find that the eigenstates are independent of the amplitude $p = \sqrt{k_x^2 + k_y^2 + 4m^2}$ and read

$$\psi_0 = \begin{pmatrix} -\sin\beta \\ \cos\beta \\ 0 \end{pmatrix}, \ \psi_+ = \begin{pmatrix} \cos\beta\,\sin\frac{\theta}{2} \\ \sin\beta\,\sin\frac{\theta}{2} \\ \cos\frac{\theta}{2}e^{i\varphi} \end{pmatrix}, \ \psi_- = \begin{pmatrix} \cos\beta\,\cos\frac{\theta}{2} \\ \sin\beta\,\cos\frac{\theta}{2} \\ -\sin\frac{\theta}{2}e^{i\varphi} \end{pmatrix}. \tag{69}$$

The wavefunction $\psi_+$ (69) is smoothly defined except at $\theta = 0$ where it is singular: it corresponds to a South gauge choice $\psi_+^S$. The North gauge can be deduced by the transformation $\psi_+^N = e^{-i\varphi}\psi_+^S$. The corresponding Berry connections are given by Eq. (40).

A simple computation leads to

$$\nu_+ = \frac{1}{2\pi}\oint_{\partial U_N} \left(\mathbf{A}_+^N - \mathbf{A}_+^S\right) \cdot d\boldsymbol{\ell} = -1. \tag{70}$$

This is expected since the 2D Berry winding $\gamma_+$ is quantized and the existence of a $\mathcal{PT}$ symmetry implies

$$\nu_+ = \frac{1}{\pi}\gamma_+. \tag{71}$$

# D   Topological properties of the Lieb model

The 2D and 3D Hamiltonians of the Lieb model are given respectively by Eqs. (42) and (44). In this Appendix we calculate their eigenstates in different gauge choices and derive the existence of the associated Dirac string. The eigenstates basis reads, in spherical coordinates $(p, \theta, \varphi)$ in $\mathbf{p} = (\mathbf{k}, m)$ space :

$$\psi_0 = \begin{pmatrix} \sin\theta \, \sin\varphi \\ \sin\theta \, \cos\varphi \\ \cos\theta \end{pmatrix}, \ \varepsilon_0 = 0 \tag{72a}$$

$$\psi_\pm = \frac{1}{\sqrt{2}} \begin{pmatrix} \cos\theta \sin\varphi \pm i \cos\varphi \\ \cos\theta \cos\varphi \mp i \sin\varphi \\ -\sin\theta \end{pmatrix}, \ \varepsilon_\pm = \pm h. \tag{72b}$$

The eigenstates only depend on $\hat{\mathbf{p}} = \mathbf{p}/p$ allowing us to focus on the unit sphere around the crossing in the 3D $\mathbf{h}$ space. Notice that the wavefunction $\psi_\pm$ (72b) for the upper and lower band is singular along the $m$-axis ($\theta = 0, \pi$), where $\varphi$ is ill-defined. For the $\varepsilon_+$ band at the North Pole we get

$$\lim_{\theta \to 0} \psi_+ = \frac{1}{\sqrt{2}} \begin{pmatrix} ie^{-i\varphi} \\ e^{-i\varphi} \\ 0 \end{pmatrix}. \tag{73}$$

Thus, the eigenstates we have considered (72b) define the South gauge $\psi_+^S$. Wavefunctions in the North gauge are defined by $\psi_+^N = e^{i\varphi}\psi_+^S$. From equation (72b) we deduce the Berry connections (46) for the upper band in the different gauges. The Chern number along a sphere $\mathcal{S}$ encircling the nodal point is readily computed,

$$\nu_+ = \frac{1}{2\pi} \oiint_{\mathcal{S}} \mathbf{F}_+ d\mathbf{S} = \frac{1}{2\pi} \left( \iint_{\mathcal{S}_N} \mathbf{F}_+ d\mathbf{S} + \iint_{\mathcal{S}_S} \mathbf{F}_+ d\mathbf{S} \right) \tag{74}$$

$$= \frac{1}{2\pi} \oint_{\theta=\pi/2} (\mathbf{A}_+^N - \mathbf{A}_+^S) d\boldsymbol{\ell} \tag{75}$$

$$= \pm 2. \tag{76}$$

Let us now show the existence of a Dirac string in the North gauge for the upper band. The derivation is identical in any gauge or band. For a given value of the angle $\theta$, consider the flux threading a disk of radius $k \sin\theta$ centered on the $m$ axis. It is given by the circulation of the Berry connection along the circle

$$\oint \mathbf{A}_+^N d\mathbf{p} = \int_0^{2\pi} d\varphi (1 - \cos\theta) = 2\pi(1 - \cos\theta). \tag{77}$$

When $\theta \to \pi$, the circle contracts to a point but the flux goes to the finite value $4\pi$. Hence the Berry connection $\mathbf{A}_+^N$ (46) describes a Berry monopole and a Dirac string carrying the flux $4\pi$ along the $m < 0$ semi-axis. This is consistent with the value of the Chern number $\nu = 2$.

# E  $\alpha$-$\mathbf{T}_3$ model

## E.1  Topological properties

Using the spherical coordinates $(p, \theta, \varphi)$ in $\mathbf{p} = (\mathbf{k}, m)$ space, the eigenstates of the Hamiltonian (55) are found to depend only on $\theta, \varphi$ for any parameter $\beta$:

$$\psi_\pm = \frac{1}{\sqrt{2(1 \pm \cos\theta \cos 2\beta)}} \begin{pmatrix} (1 \pm \cos\theta) \cos\beta \, e^{-i\varphi} \\ (1 \mp \cos\theta) \sin\beta \, e^{i\varphi} \\ \pm \sin\theta \end{pmatrix} , \quad \varepsilon_\pm = \pm\sqrt{k^2 + m^2}, \tag{78a}$$

$$\psi_0 = \frac{1}{\sqrt{1 - \cos^2\theta \cos^2 2\beta}} \begin{pmatrix} -\sin\theta \sin\beta \, e^{-i\varphi} \\ \sin\theta \cos\beta \, e^{i\varphi} \\ \cos\theta \sin 2\beta \end{pmatrix} , \quad \varepsilon_0 = -m\cos 2\beta, \tag{78b}$$

Let us now focus on the upper band for illustration: the wavefunction (78a) has singularities at the North pole $\theta = 0$ and the South Pole $\theta = \pi$. We can regularize it at the North (resp. South) pole through $\psi_+^N = e^{i\varphi}\psi_+$ (resp. $\psi_+^S = e^{-i\varphi}\psi_+$) The wavefunction $\psi_+^N$ ($\psi_+^S$) has a unique vortex at the South (North) pole and is smoothly defined elsewhere. The associated Berry connections for the $\varepsilon_+$ band read

$$\mathbf{A}_+^\mathbf{N} = \frac{\sin^2\theta + 2(1 - \cos\theta)^2 \sin^2\beta}{2(1 + \cos\theta \cos 2\beta)} \nabla_\mathbf{p}\varphi, \tag{79}$$

$$\mathbf{A}_+^\mathbf{S} = -\frac{\sin^2\theta + 2(1 + \cos\theta)^2 \cos^2\beta}{2(1 + \cos\theta \cos 2\beta)} \nabla_\mathbf{p}\varphi. \tag{80}$$

As in the magnetic monopole case, these connections describe a source of Berry flux at the origin together with a half flux tube on the $m < 0$ (resp. $m > 0$) semi-axis, see Fig. 2. These half flux are determined by considering the winding of the connection around a circle of radius $k \sin\theta$ at polar angle $\theta$:

$$\oint_\theta \mathbf{A}_+^\mathbf{N} d\mathbf{p} = \int_0^{2\pi} \frac{\sin^2\theta + 2(1 - \cos\theta)^2 \sin^2\beta}{2(1 + \cos\theta \cos 2\beta)} d\varphi$$
$$= \pi \frac{(\sin^2\theta + 2(1 - \cos\theta)^2 \sin^2\beta)}{(1 + \cos\theta \cos 2\beta)}. \tag{81}$$

Equation (81) contains the flux from the Berry monopole located at the origin through the surface of the disk and a possible contribution from the half-flux tube. The first contribution increases with the solid angle of the surface. In the limit $\theta \to 0$ (North pole) or $\theta \to \pi$ (South pole) we get an extra contribution:

$$\lim_{\theta \to 0} \oint \mathbf{A}_+^\mathbf{N} d\mathbf{k} = 0, \quad \lim_{\theta \to \pi} \oint \mathbf{A}_+^\mathbf{N} d\mathbf{k} = \frac{8\pi \sin^2\beta}{1 - \cos 2\beta}, \tag{82}$$

which corresponds to the Berry flux carried by the Dirac half string. Similarly, we find that the South connection $\mathbf{A}_+^S$ describes a half-flux tube located on the positive $m$ semi-axis

$$\lim_{\theta \to 0} \oint \mathbf{A}_+^\mathbf{S} d\mathbf{p} = -\frac{8\pi \cos^2\beta}{1 + \cos 2\beta}, \quad \lim_{\theta \to \pi} \oint \mathbf{A}_+^\mathbf{N} d\mathbf{p} = 0. \tag{83}$$

For $\beta = \pi/4$ the difference $\mathbf{A}_+^N - \mathbf{A}_+^S$, corresponding to the gauge transformation $\psi \to e^{2i\varphi}\psi$, describes an infinite solenoid of flux $4\pi\hat{e}_z = 2\pi\nu\hat{e}_z$, where $\nu$ is the Chern number associated to the upper band. When $\beta \neq \pi/4$, the flux is not quantized in units of $2\pi$. This non-quantization reflects the non quantization of the 2D Berry windings (51) in units of $\pi$.

Finally, the Chern number of the $\epsilon_+$ gapped band reads:

$$
\begin{aligned}
\nu_+ = \frac{1}{2\pi} \oiint_{\mathcal{S}} \mathbf{F}_+ d\mathbf{S} &= \frac{1}{2\pi} \left( \iint_{\mathcal{S}_N} \mathbf{F}_+ d\mathbf{S} + \iint_{\mathcal{S}_S} \mathbf{F}_+ d\mathbf{S} \right) \\
&= \frac{1}{2\pi} \oint_{m=0} (\mathbf{A}_+^N - \mathbf{A}_+^S) d\mathbf{k} \\
&= 2.
\end{aligned}
\tag{84}
$$

Note that the Chern number is independent of $\beta$, a manifestation of its topological nature and the absence of gap closing for $\beta \neq 0, \pm\pi/2$.

## E.2 A realisation of the $\alpha$-T$_3$ model in critical HgCdTe

The three-band crossing in critical HgCdTe can be described by the linear $\mathbf{k} \cdot \mathbf{p}$ Hamiltonian of the Kane model [20]. It describes the band structure of Zinc-blende semiconductors at the $\Gamma$ point. The conduction band has orbital degeneracy 1 and is $s$-type ($|u_s\rangle$) whereas the valence band is $p$-type ($|u_x\rangle, |u_y\rangle, |u_z\rangle$). From the atomic-like states of the valence band one forms the following basis of eigenfunctions of the total angular momentum projection on the $z$-axis: the band is split into two subspaces of total angular momentum $J = 1/2$ and $J = 3/2$, the first manifold being set far down under the topmost valence band because of spin-orbit coupling: $E(J = 1/2) \ll E(J = 3/2), E(|u_s\rangle)$. In the new basis ($|u_s, \uparrow\rangle, |u_{3/2, +3/2}\rangle, |u_{3/2, -1/2}\rangle, |u_s, \downarrow\rangle, |u_{3/2, -3/2}\rangle, |u_{3/2, +1/2}\rangle$) the low-energy Hamiltonian reads

$$
H(\mathbf{k}) = \begin{pmatrix}
0 & \frac{\sqrt{3}}{2}k_- & 0 & 0 & 0 & 0 \\
\frac{\sqrt{3}}{2}k_+ & 0 & -\frac{k_-}{2} & -m & 0 & 0 \\
0 & -\frac{k_+}{2} & 0 & 0 & -m & 0 \\
0 & -m & 0 & 0 & \frac{k_-}{2} & 0 \\
0 & 0 & -m & \frac{k_+}{2} & 0 & -\frac{\sqrt{3}}{2}k_- \\
0 & 0 & 0 & 0 & -\frac{\sqrt{3}}{2}k_+ & 0
\end{pmatrix},
\tag{85}
$$

where we used the notation $k_\pm = k_x \pm i k_y$ for compactness. The spectrum is given by $\varepsilon = 0, \pm k$, each energy level being doubly degenerate. An eigenstate basis is given by:

$$\psi_A^\pm = \frac{1}{\sqrt{2}} \begin{pmatrix} \frac{\sqrt{3}}{2} e^{-i\varphi} \sin\theta \\ \pm 1 \\ -\frac{1}{2} e^{i\varphi} \sin\theta \\ -\cos\theta \\ 0 \\ 0 \end{pmatrix} \quad , \quad \psi_B^\pm = \frac{1}{\sqrt{2}} \begin{pmatrix} 0 \\ 0 \\ \cos\theta \\ -\frac{1}{2} e^{-i\varphi} \sin\theta \\ \mp 1 \\ \frac{\sqrt{3}}{2} e^{i\varphi} \sin\theta \end{pmatrix} \quad , \quad \varepsilon_\pm = \pm k, \tag{86a}$$

$$\psi_A^0 = \begin{pmatrix} \frac{1}{2} e^{-3i\varphi} \sin\theta \\ 0 \\ \frac{\sqrt{3}}{2} e^{-i\varphi} \sin\theta \\ 0 \\ 0 \\ -\cos\theta \end{pmatrix} \quad , \quad \psi_B^0 = \begin{pmatrix} \cos\theta \\ 0 \\ 0 \\ \frac{\sqrt{3}}{2} e^{i\varphi} \sin\theta \\ 0 \\ \frac{1}{2} e^{3i\varphi} \sin\theta \end{pmatrix} \quad , \quad \varepsilon_0 = 0, \tag{86b}$$

where we have used spherical coordinates around the degeneracy point $(k, \theta, \varphi)$. Note that these wavefunctions do not exhibit any vortex or phase winding: in particular their phase is well defined at the poles. Hence, the Chern number must be zero for any band around the crossing. This absence of topological protection can be expected since the crossing is achieved by fine tuning of the Cd concentration [20].

Nevertheless, at the equator $m = 0$ ($\theta = \pi/2$) the bands exhibit a non-zero Berry winding. Indeed at the equator the Hamiltonian (85) becomes block diagonal, where each block A/B corresponds to one valley of the $\alpha$-$T_3$ model for $\tan\beta = \alpha = \frac{1}{\sqrt{3}}$ [17, 21]. The corresponding windings are

$$\gamma_\xi^\pm = -\xi \frac{\pi}{2} \ , \ \gamma_\xi^0 = -\xi\, 3\pi = \pi \bmod 2\pi, \tag{87}$$

where $\xi = \pm 1$ for the A/B sector.

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
