# Peer review of "Quantized Berry winding from an emergent $\mathcal{PT}$ symmetry"

_SciPost Physics_

## Round 1 · Referee Report · Anonymous (Referee 1) · 2022-1-3

Strengths

1 – The authors succeed to relate the quantization of the Berry winding in D=2 to the quantization of a Chern number on a surface in D=3 using an underlying symmetry.
2 – The distinction between Berry phase/Wilson loop and Berry winding is conceptually insightful.
3 – The introduced models built onto each other and they are chosen to discuss meaningful limiting cases.

Weaknesses

1 – Besides graphene no material example is mentioned for such a quantized Berry winding due to $\mathcal{PT}$ symmetry. The example of HgCdTe is stated to have a non-quantized Berry winding.
2 – While for the connection to the Chern numbers the distinction between Berry phase and Berry winding is necessary, it remains unclear whether this has further implications.

Report

The manuscript “Quantized Berry winding from an emergent PT symmetry” by Louvet et al. discusses a relation between the Berry winding in a two-dimensional semimetal and the Chern number of a point crossing embedded in three dimensions. Embedding means the introduction of a mass term that gaps out the original band crossing, whereby the mass becomes an additional dimension. In the presence of a $\mathcal{PT}$ symmetry, which relates eigenstates that are smooth on different open sets, the Berry winding is found to be quantized. The authors generalize their embedding into 3D to n-band crossings, e.g., for Hamiltonians constructed from arbitrarily sized spin matrices. With this method they consider three three-band models and study the relations between Berry winding and Chern numbers.
Overall the work is worth publishing, but it seems questionable whether the impact requirements of SciPost Physics are met. Below some specific points are addressed how this conclusion is reached and suggestions are given in which directions the work might be extended to be recommended for publication.
First some general remarks on related ideas to review the extend of originality. After embedding a 2D band crossing into a 3D parameter space, the authors’ general process follows the discussion of magnetic monopoles and their Dirac strings: The magnetic vector potential cannot be smoothly defined on a sphere enclosing the monopole. Therefore two different gauges must be used describing the northern and southern hemisphere, for which then Stokes theorem applies, see for example Ch. 1.9 in Nakahara, Mikio. Geometry, topology and physics. CRC press, 2018. For point crossings as sources of Berry curvature analogous arguments can be made by replacing the magnetic field and vector potential by the Berry curvature and the Berry connection. Since the wave functions must be continuous when encircling the band crossing, there is a restriction on the relation between the two different gauges mentioned before. As a result one finds that the Chern number, which can be written as the difference of gauge-dependent Berry winding at the boundary of the hemispheres is quantized to an integer, see for example Kohmoto, Mahito. Annals of Physics 160.2 (1985): 343-354. So far this quantization is valid without symmetry, but one does not obtain any information about the Berry winding for each gauge on its own, i.e., only for the total value.
The present paper goes a step further and introduces another symmetry, that fixes the relation between the two gauges of the Berry connection. Since the total of the two Berry windings is fixed to the Chern number and their relation is now given, it becomes possible to obtain the winding itself. The quantization of a Berry phase due to mirror, inversion or PT symmetries has been discussed before, see J. Zak Phys. Rev. Lett. 62, 2747 (1989) and Ref.12 (arXiv:1810.04094). But whereas previous approaches have considered crystalline symmetries that quantize the Berry phase, here Louvet et al. consider a PT symmetry that is only emergent in the low-energy description in the vicinity of a band crossing.
To summarize this comparison, their approach brings a new twist to old ideas. My main concern is whether this emergent PT symmetry is a widespread feature or only due to the fine-tuning of the considered models. The latter would limit from my perspective the impact. When the authors discuss the difference to Ref.12, it is stated that their emergent symmetry can also be used without an underlying crystal. To find a new pathway for future research, I would deem it necessary that this extension becomes clearer. Can materials/systems be proposed which are sufficiently close to this emergent symmetry? What “continuous media” do the authors have in mind?

I failed to confirm that the states in Eq. 24 are eigenstates of the Hamiltonian given in Eq. 23. Am I correct that these states are supposed to be eigenstates?
Regarding the paragraph after Eq. 24: Could the authors explain in more detail how they define the Berry connection for this orientation of the Dirac string? Are they considering not $A_\pm = 0$ but some distribution that is able to describe the discontinuity in the wave functions and gives a finite contribution in the integration?

In the conclusions it is stated that a “[…] necessary condition for the Berry winding … to be quantized.” has been discussed. It appears the word “necessary” is used here for the first time. Does this mean that if the Berry phase is quantized due to inversion or mirror symmetries, one will also find an emergent PT symmetry? Or is the condition rather “sufficient” than “necessary”?

One possibility to extend the paper could be to consider the “Berry winding” in more depth and how it differs from the Berry phase, which, if quantized, only takes the values 0 or $\pi$. Maybe it can be elaborated in more detail what the connection to the minimal conductivity mentioned in the context of Ref. 18 would be. After Eq. 48 it is pointed out that a Berry winding of $2\pi$ is different to a winding of 0, does this have effects on the results of Ref. 18? Is there a way to introduce higher Berry winding numbers to possibly raise the minimal conductivity at the crossing further?

Some additional remarks and questions:
A source of possible confusion might be calling symmetries like V in Eq. 16 at PT symmetry, because they mirror the m variable. Maybe calling them C2*T would be clearer, having the same interpretation in 2D and 3D.
In Eq. 2 it seems to me that the signs are not consistent, is the angle phi measured clock-wise or counter-clock-wise? If the latter is true I get the opposite sign for the right hand-side of Eq., please have a look.
In Eqs. 7A and 7b a different sign convention seems to be chosen than in Eq.2, since the minus is absent in Eq.19 I suspect it might be a typo in front of Im <… .
Is the second half of Eq. 32 correct? As far as I understand the symmetry V should leave k invariant, yet there is a sign change for $k_y$ to $-k_y$, which I also could not reproduce.
In Eq. 52 I would expect a commutation relation, but then the order of operations is not reversed on either side of the equalities.
In Eq. 60 it seems that there is complex unit missing, since the log W should be equal to “i Arg W” if W is only a number.

Requested changes

1 - Besides the points addressed in the main report to extend the scope of the manuscript, I would suggest to add some more references to discuss related works. Some literature on Dirac strings and the whole business of choosing a smooth gauge and a reference for the first paragraph in appendix B would be helpful.

2- In Fig. 2 the blue lines are not mentioned in the description and it seems confusing that the origin of the coordinate system, labeled with 0, does not coincide with the red sphere, which I suspect to be the band crossing point, which is said to be at k = 0.

3 - Minor typos/suggestions: - In the introduction PT is only partially formatted as $\mathcal{PT}$ - When introducing the PT symmetry, one might add definitions of T and P by themselves. - Before Eq. 3 it says “along a close path”, it is not quite clear what it is “close” to or should it say “closed”? - In Eq. 3 I suspect that the third term should also have a $\pm$. - Eigenstates $\psi_+^N \ldots$ are written with and without kets, maybe this can be unified. - Caption Fig. 1: “dimenisonal” → “dimension”; union of the hemispheres should have the union symbol instead of “U” - “obstruction to smoothly define smoothly eigenstates” drop one “smoothly” - “p-sphere” sounds like the term “n-sphere” which is likely not meant here? - Is the reference to Fig. 2 after Eq. 9 is possibly meant to point towards Fig.1? - The sentence after Eq. 31 is unclear, are $\theta$ and $\varphi$ not spherical coordinates? - There are two occasion of “connexion” instead of “connection” - After Eq. 39 in “by introducing the the vector” there is one “the” too many. - Before Eq. 54 it says once “Chern numnber” - In “given than no gap closes” I suspect “than” → “that” - Before Eq.61 “subsset” → “subset”

  • validity: ok
  • significance: low
  • originality: low
  • clarity: good
  • formatting: good
  • grammar: reasonable

Author:  Thibaud Louvet  on 2023-03-08  [id 3449]

(in reply to Report 1 on 2022-01-03)

Report 1

Dear referee, We apologize for the long delay of our response, due to personal reasons in this post-covid period. We have substantially modified our manuscript taking into account the reports.

We thank the referee for noticing that “The authors succeed to relate the quantization of the Berry winding [...] using an underlying symmetry”, in acknowledging that “The distinction between Berry phase/Wilson loop and Berry winding is conceptually insightful” and that “The introduced models [...] are chosen to discuss meaningful limiting cases.”

Weaknesses The referee noticed several weaknesses in his report, that we address below.

1."Besides graphene no material example is mentioned for such a quantized Berry winding due to PT symmetry. The example of HgCdTe is stated to have a non-quantized Berry winding." Answer: Indeed, in two dimensional crystals, the simplest and most common example of a linear crossing is a Dirac cone. Such a Dirac cone occurs in graphene, but also at the surface of D=3 Topological Insulators. We have included in the introduction a review of Dirac cone realisations beyond graphene and beyond electronic crystals. Indeed, we stress that our approach is not limited to crystalline materials: we have added an example from the physics of fluids in geophysics, the shallow water model, where a spin-1 band crossing between 3 bands emerges. Such a model possesses a quantized Berry winding related to a finite Chern number through a $PT$ symmetry local in momentum. Modification: p.2 section 1 we have added a short review of Dirac cones realizations beyond graphene. p.15 section 3.2 we have added a discussion of the shallow water model, a physical realization of the spin-1 band crossing protected by $PT$ symmetry.

2."While for the connection to the Chern numbers the distinction between Berry phase and Berry winding is necessary, it remains unclear whether this has further implications." Answer: We do agree that in the existing literature on band crossings the Berry/Zak phase is usually considered. Besides our analysis on the relation between a Berry winding and a Chern number, we note that the difference between Berry/Zak phase and Berry winding has several physical consequences. In the context of the Quantum Hall Effect, the semi-classical quantization of the Landau levels relies on the Berry winding and not the Berry phase. On the contrary, we have shown in a previous work that the minimal conductivity reached for a chemical potential at a band crossing vanishes for models without a quantized Berry phase [Th. Louvet et al., Phys. Rev. B 92, 155116 (2015)]. This suggests a relation between the nature of the evanescent states of a semimetallic ribbon and the Berry phase of the bulk dispersion relation. Furthermore it has also been shown that the suppression of backscattering and other interference effects in Dirac materials is related to the Berry phase: interferences are by essence a "phase effect" [T. Ando et al., J. Phys. Soc. Jpn. 67, pp. 2857-2862 (1998)]. Modification: p.3 section 2.1 we have added a discussion of the different physical implications of the Berry winding and the Berry phase.

Report:

We acknowledge that referee finds that "Overall the work is worth publishing, but it seems questionable whether the impact requirements of SciPost Physics are met. Below some specific points are addressed how this conclusion is reached and suggestions are given in which directions the work might be extended to be recommended for publication."

1.The referee then provides a very thorough summary of our approach : "First some general remarks on related ideas to review the extend of originality. After embedding a 2D band crossing into a 3D parameter space, the authors general process follows the discussion of magnetic monopoles and their Dirac strings: The magnetic vector potential cannot be smoothly defined on a sphere enclosing the monopole. Therefore two different gauges must be used describing the northern and southern hemisphere, for which then Stokes theorem applies, see for example Ch. 1.9 in Nakahara, Mikio. Geometry, topology and physics. CRC press, 2018. For point crossings as sources of Berry curvature analogous arguments can be made by replacing the magnetic field and vector potential by the Berry curvature and the Berry connection. Since the wave functions must be continuous when encircling the band crossing, there is a restriction on the relation between the two different gauges mentioned before. As a result one finds that the Chern number, which can be written as the difference of gauge-dependent Berry winding at the boundary of the hemispheres is quantized to an integer, see for example Kohmoto, Mahito. Annals of Physics 160.2 (1985): 343-354. So far this quantization is valid without symmetry, but one does not obtain any information about the Berry winding for each gauge on its own, i.e., only for the total value. The present paper goes a step further and introduces another symmetry, that fixes the relation between the two gauges of the Berry connection. Since the total of the two Berry windings is fixed to the Chern number and their relation is now given, it becomes possible to obtain the winding itself. The quantization of a Berry phase due to mirror, inversion or PT symmetries has been discussed before, see J. Zak Phys. Rev. Lett. 62, 2747 (1989) and Ref.12 (arXiv:1810.04094). But whereas previous approaches have considered crystalline symmetries that quantize the Berry phase, here Louvet et al. consider a PT symmetry that is only emergent in the low-energy description in the vicinity of a band crossing. To summarize this comparison, their approach brings a new twist to old ideas. My main concern is whether this emergent PT symmetry is a widespread feature or only due to the fine-tuning of the considered models. The latter would limit from my perspective the impact." Answer: We intentionally focused on properties of the low energy dispersion, specific to the intrinsic nature of the band crossing irrespective of a specific material symmetries. We indeed believe that the PT symmetry is an ubiquitous emerging symmetry of generic band crossings. It is a symmetry local in momentum, and relating positive and negative energy bands around the band crossing, and therefore it does not presume any momentum space structure associated to crystalline symmetries.
Our approach is targeted towards the understanding of topological properties of occurrences of band crossings in any material, crystalline or not. Indeed the referee is right to note that it is possible to have a system where PT symmetry emerges at low energy and is broken at higher energy. While the band crossing itself may require some fine tuning, as suggested by the referee, the PT symmetry is a property of the type of band crossing and is rather generic.

2."When the authors discuss the difference to Ref.12, it is stated that their emergent symmetry can also be used without an underlying crystal. To find a new pathway for future research, I would deem it necessary that this extension becomes clearer. Can materials/systems be proposed which are sufficiently close to this emergent symmetry? What continuous media do the authors have in mind?" Answer: Waves in geo/astrophysical flows, but also magnetized plasmas, chiral active fluids and gyrotropic media have been recently shown to exhibit topological properties that can be captured by a Berry monopole in three-dimensional parameter space (Delplace et al. Science 358, 1075 (2017), Souslov et al. Phys. Rev. Lett. 122, 128001 (2019), Mechelen and Jacob, Phys. Rev. A 98, 023842 (2018), Marciani and Delplace Phys. Rev. A 101, 023827 (2020)). Furthermore, the models derived in these works to describe the topology of such continuous systems turn out to show a pseudo-spin 1 structure. They are therefore automatically captured by our analysis. Modifications: In the new version of the manuscript, we illustrate the use of our approach on a concrete continuous model, the rotating shallow water model, a textbook fluid model describing the dynamics of a thin layer of incompressible fluid in a rotating frame. This model is commonly used to describe oceanic and atmospheric waves over planetary scales. In its linear version, the rotating shallow water model reduces to a pseudo-spin 1 Hamiltonian with a three-dimensional parameter space $(k_x,k_y,f)$, where $f$, called the Coriolis parameter, plays the role of the mass term in our analysis. Actually, a discretized version of the rotating shallow water model was previously shown to coincide with a Lieb lattice model with imaginary hopping terms to the second nearest neighbors. We have thus added the discussion on this continuous model at the end of the section 3.2 that deals with the Lieb lattice.

3."I failed to confirm that the states in Eq. 24 are eigenstates of the Hamiltonian given in Eq. 23. Am I correct that these states are supposed to be eigenstates?" Answer: We thank the referee for pointing out a mistake in the text. There was a typo, and the correct definition of $\varphi$ in this equation is $\varphi = \mathrm{arctan}(k_x/k_y)$ instead of $k_y/k_x$. Modification: p.8 section 2.4 we have corrected the definition of $\varphi$.

4."Regarding the paragraph after Eq. 24: Could the authors explain in more detail how they define the Berry connection for this orientation of the Dirac string? Are they considering not $A_\pm = 0$ but some distribution that is able to describe the discontinuity in the wave functions and gives a finite contribution in the integration?" Answer: It is mentioned in Berry's original paper that if the function is not single-valuated along the loop then the integral of the Berry connection is ill-defined. It is the case here: the Berry connection is zero everywhere around the closed loop surrounding the crossing except at the point where the function is multi-valuated, where the Berry connection is ill-defined. As noted by Berry, embedding the closed loop in 3D and considering a surface bordering the loop lifts this ambiguity. Modification:p.8 section 2.4 we have clarified the derivation of the Berry winding and added a discussion of Berry's original remark.

5."In the conclusions it is stated that a "necessary condition for the Berry winding to be quantized has been discussed". It appears the word necessary is used here for the first time. Does this mean that if the Berry phase is quantized due to inversion or mirror symmetries, one will also find an emergent PT symmetry? Or is the condition rather sufficient than necessary?" Answer: We thank the referee for pointing out this wrong wording. The condition is indeed "sufficient" and not "necessary", as is mentioned elsewhere in the text (e.g. after Eq.19). We have corrected this mistake. Modification: p.17 section 4 we have replaced "necessary" by "sufficient".

6."One possibility to extend the paper could be to consider the Berry winding in more depth and how it differs from the Berry phase, which, if quantized, only takes the values $0$ or $\pi$. Maybe it can be elaborated in more detail what the connection to the minimal conductivity mentioned in the context of Ref. 18 would be. After Eq. 48 it is pointed out that a Berry winding of $2\pi$ is different to a winding of $0$, does this have effects on the results of Ref. 18? Is there a way to introduce higher Berry winding numbers to possibly raise the minimal conductivity at the crossing further?" Answer: We thank the referee for pointing out this line of development. We have elaborated on the differences between the Berry phase and Berry winding in section 2.1. We have added a discussion of the implication of the difference between Berry winding and Berry phase based on the results of [Th. Louvet et al., Phys. Rev. B 92, 155116 (2015)]. The correlation found was between the minimal conductivity and the Berry phase. We stress that contrary to the Berry winding, the Berry phase is only defined modulo $2\pi$. If the Berry winding was at the origin of the minimal conductivity, we could have hoped indeed that higher Berry winding numbers would raise the minimal conductivity; but since it is the Berry phase, we could on the contrary conjecture that the value of the minimal conductivity should remain the same, a "universal" finite minimal conductivity for crossings with finite Berry phase (odd Berry winding numbers). \bf Modification: p.3 section 2.1 we have clarified the difference between Berry winding and Berry phase and proposed a conjecture based on previous results of [Th. Louvet {\it et al.}, Phys. Rev. B 92, 155116 (2015)].

Some additional remarks and questions:

1."A source of possible confusion might be calling symmetries like $V$ in Eq. 16 a $\mathcal{PT}$ symmetry, because they mirror the m variable. Maybe calling them $C_2\mathcal{T}$ would be clearer, having the same interpretation in 2D and 3D. In Eq. 2 it seems to me that the signs are not consistent, is the angle phi measured clock-wise or counter-clock-wise? If the latter is true I get the opposite sign for the right hand-side of Eq. ?, please have a look." Answer: We do agree with the referee that the symmetry acts only, like any symmetry, on the 2D real/momentum space of the model, and thus not on the parameter space associated with the mass $m$. However we believe that this is a rather standard convention, and that denoting an antinunitary symmetry local in momentum a $C_2\mathcal{T}$ rather than $P\mathcal{T}$ would add more confusion than it would help. Therefore we prefer to stick to our initial and standard naming. The angle $\varphi$ is defined by: $(k_x,k_y)=k(\cos\varphi,\sin\varphi)$. After verification, we believe that the signs of Eq.(2) seem correct. We recall that $\psi_+ = (1,\mathrm{e}^{i\varphi})$.

2."In Eqs. 7A and 7b a different sign convention seems to be chosen than in Eq.2, since the minus is absent in Eq.19 I suspect it might be a typo in front of $\mathrm{Im} \langle...$." Answer: There was indeed a sign error in eqs. (7a) and (7b), we thank the referee for pointing it out. The sign convention is the same, $\vec{A} = -i \langle \psi | \vec{\nabla} | \psi \rangle = \mathrm{Im} \langle \psi | \vec{\nabla} | \psi \rangle$. \ {\bf Modification:} p.5 sections 2.2 eqs.(7a) and (7b) we have corrected the sign error.

3."Is the second half of Eq. 32 correct? As far as I understand the symmetry V should leave k invariant, yet there is a sign change for $k_y$ to $-k_y$, which I also could not reproduce." Answer: The referee mentions the following equation, p.10 in section 2.5.2: $V H^{3D} (\mathbf{k},m) V^{-1} = H^{3D} (\mathbf{k},- m) ~\rightarrow~ U H^{3D} (k_x,k_y,m) U^{-1} = H^{3D} (k_x,- k_y, -m).$ This equation describes the action of the $PT$ symmetry $V$ on the Hamiltonian. The action of $V$ does leave the momentum $\bf k$ invariant. The second part of this equation describes how the operator $U$ acts on the Hamiltonian. Knowing that $V=U \mathcal{K}$ with $\mathcal{K}$ the complex conjugation operator and $H = S_x k_x + S_y k_y + S_z k_z$ with matrices $S_x$, $S_z$ being real and matrix $S_y$ being purely imaginary, we find that $U$ necesarily flips $k_y$. In a nutshell, $V$ does leave $k_y$ invariant while $U$ reverses its sign.

4."In Eq. 52 I would expect a commutation relation, but then the order of operations is not reversed on either side of the equalities." Answer: Thanks for pointing out this typo. Modification: p.16 section 3.3 we have replaced $\Sigma_1 U = \Sigma_1 U, \qquad \Sigma_2 U = - \Sigma_2 U$ by the correct commutations relations $\Sigma_1 U = U \Sigma_1, \qquad \Sigma_2 U = - U \Sigma_2$.

5."In Eq. 60 it seems that there is complex unit missing, since the log W should be equal to i Arg W if W is only a number." Answer: Thank you, we have corrected the typo. Modification: p.19 appendix A the relation between Wilson loop and Berry winding now reads as it should: $\mathrm{i} \log W [ \ell ] = \pi \gamma_\ell ~\mathrm{mod}~ 2\pi$. We have added the complex unit $\mathrm{i}$ that was missing in the previous version.

Requested changes

1."Besides the points addressed in the main report to extend the scope of the manuscript, I would suggest to add some more references to discuss related works. Some literature on Dirac strings and the whole business of choosing a smooth gauge and a reference for the first paragraph in appendix B would be helpful." Answer: We agree with the referee. We have added related references. Modifications: p.4 section 2.2 we have added historical references to the 1931 Dirac paper, the 1975 Wu and Yang paper, the 1985 Kohmoto paper as well as general references: the 1990 book by Nakahara and a recent review paper by Cayssol and Fuchs (2021). p.19 appendix B we have added a reference to equation (44) for clarity.

2."In Fig. 2 the blue lines are not mentioned in the description and it seems confusing that the origin of the coordinate system, labeled with 0, does not coincide with the red sphere, which I suspect to be the band crossing point, which is said to be at k = 0." Answer: The referee mentions the Figure whose caption begins with "A 2D band crossing in momentum space", now numbered Figure 1. We agree with the referee's remarks and have modified the figure and caption accordingly. Modification: p.5 section 2.2, Figure 1: we have removed the $0$ label from the coordinate axes since $k=0$ corresponds to the red sphere. We have added a description of the blue arrows in the caption: they illustrate the Berry flux.

3.We have corrected the list of typos noted by the Referee : "- In the introduction PT is only partially formatted as $\mathcal{PT}$ - When introducing the PT symmetry, one might add definitions of $\mathcal{T}$ and $\mathcal{P}$ by themselves. - Before Eq. 3 it says along a close path, it is not quite clear what it is close to or should it say closed? - In Eq. 3 I suspect that the third term should also have a $\pm$. - Eigenstates $\psi_N^+$ are written with and without kets, maybe this can be unified. - Caption Fig. 1: dimenisonal $\to$ dimension; union of the hemispheres should have the union symbol instead of U - obstruction to smoothly define smoothly eigenstates drop one smoothly - p-sphere sounds like the term n-sphere which is likely not meant here? - Is the reference to Fig. 2 after Eq. 9 is possibly meant to point towards Fig.1? - The sentence after Eq. 31 is unclear, are $\theta$ and $\varphi$ not spherical coordinates? - There are two occasion of connexion instead of connection - After Eq. 39 in by introducing the the vector there is one the too many. - Before Eq. 54 it says once Chern numnber - In given than no gap closes I suspect than $\to$ that - Before Eq.61 subsset $\to$ subset "

Please find attached a 'diff.pdf' file showing changes.

With our best regards, Thibaud Louvet, Pierre Delplace, Mark Oliver Goerbig and David Carpentier

Attachment:

diff_0k589R7.pdf

---

## Round 1 · Referee Report · Anonymous (Referee 2) · 2022-1-19

Report

Comments:

  1. Paragraph after Eq. 19: “Here we show that an effective PT symmetry is a sufficient condition for the quantization.” It is necessary not sufficient condition. This is also what is written in the first sentence in the Conclusions.

  2. Eq 45, I believe it is meant strictly for SU(2) algebra. As it’s written, it can be any Lie algebra, e.g., for SU(3) the Einstein summation is assumed for 8 dimensional basis. But here it is the case of spin-1 (i.e., 3*3 representation of SU(2) algebra).

  3. It is confusing to first read that in Sec 2.5 for n-band crossing, the procedure works following the same spirit for 2-band crossing with the generalized formulations eqs (27-29). But it is misleading because in Sec 3.3, as a counter example given by \alpha-T3 model, such a procedure in constructing the operator (27) cannot work. What is the reason behind? The authors should clarify the shortcoming of Sec 2.5. I think it maybe closer to the special property of SU(2) matrices where the procedure works.

  4. In calculating Chern number, the domain of integration is always a two-dimensional closed manifold, simply connected (the referee is simply stating something far from the rigor of mathematics). What the actual two integrating variables parameterizing the manifold are secondary (even though whether the Chern number corresponds to physical observable or not is another question). So, I think with Chern number being so common understanding now, it is really confusing unnecessarily to call D=3 Chern number. The work proposed to take an additional "mass" parameter in addition to the two momenta variables (in two spatial dimensions), and restricting them to form a S2 sphere. This is quite well known in the literature of Weyl semimetals. And there is no such topological quantity as Chern number in 3 dimensions – as far as the standard topological classification is concerned. And I believe the authors are not doing anything different from the standard classification. Please correct the unnecessary wording.

  • validity: ok
  • significance: ok
  • originality: ok
  • clarity: ok
  • formatting: reasonable
  • grammar: excellent

Author:  Thibaud Louvet  on 2023-03-08  [id 3448]

(in reply to Report 2 on 2022-01-19)

**Report 2**

We apologize for the long delay of our response, due to personal reasons in this post-covid period.
We have substantially modified our manuscript taking into account the reports.

1."Paragraph after Eq. 19: Here we show that an effective PT symmetry is a sufficient condition for the quantization. It is necessary not sufficient condition. This is also what is written in the first sentence in the Conclusions."
**Answer:**
On the contrary, the PT symmetry is a sufficient condition for the quantization. There was a typo in the conclusion, it has been corrected and now reads: "In this article, we have discussed a sufficient condition...".

2."Eq 45, I believe it is meant strictly for SU(2) algebra. As it's written, it can be any Lie algebra, e.g., for SU(3) the Einstein summation is assumed for 8 dimensional basis. But here it is the case of spin-1 (i.e., $3*3$ representation of SU(2) algebra)."
**Answer:**
Indeed, the referee is perfectly correct.
We have corrected the main text and written the three commutation relations explicitly, section 3.2 page 14.

3."It is confusing to first read that in Sec 2.5 for n-band crossing, the procedure works following the same spirit for 2-band crossing with the generalized formulations eqs (27-29). But it is misleading because in Sec 3.3, as a counter example given by $\alpha$-T$_3$ model, such a procedure in constructing the operator (27) cannot work. What is the reason behind? The authors should clarify the shortcoming of Sec 2.5. I think it maybe closer to the special property of SU(2) matrices where the procedure works."
**Answer:**
The procedure is the same for n-band crossing: If a PT symmetry exists, the Berry phase is quantized. In the $\alpha$-T$_3$ model, the Berry phase is not quantized, therefore it is not so surprising that no PT symmetry is found and no $V$ operator. We have added a sentence "When such an operator $V$ exists," in section 2.5 page 9 to lift the ambiguity.

4."In calculating Chern number, the domain of integration is always a two-dimensional closed manifold, simply connected (the referee is simply stating something far from the rigor of mathematics). What the actual two integrating variables parameterizing the manifold are secondary (even though whether the Chern number corresponds to physical observable or not is another question). So, I think with Chern number being so common understanding now, it is really confusing unnecessarily to call D=3 Chern number. The work proposed to take an additional "mass" parameter in addition to the two momenta variables (in two spatial dimensions), and restricting them to form a S2 sphere. This is quite well known in the literature of Weyl semimetals. And there is no such topological quantity as Chern number in 3 dimensions as far as the standard topological classification is concerned. And I believe the authors are not doing anything different from the standard classification. Please correct the unnecessary wording."
**Answer:** We have removed the expression "3D Chern number" to avoid any confusion.
We acknowledge that is was very confusing.
We have added appropriate citations and clarified when we followed the route of the usual derivation of Chern number as in the works of Dirac (1931), Kohmoto (1985) and the book of Nakahara (1990).
**Modification:** p.4 section 2.2 we have specified that we follow a standard derivation of the Chern number and added references.

Please find attached a 'diff.tex' file showing changes.

With our best regards,
Thibaud Louvet,
Pierre Delplace,
Mark Oliver Goerbig and
David Carpentier

Attachment:

diff.pdf

---

## Editorial Decision

resubmitted